# Identifying an optimal dihydroartemisinin-piperaquine dosing regimen for malaria prevention in young Ugandan children

Erika Wallender [1], Ali Mohamed Ali[2], Emma Hughes[2], Abel Kakuru[3], Prasanna Jagannathan [4], Mary Kakuru Muhindo[3], Bishop Opira[3], Meghan Whalen[1], Liusheng Huang [1], Marvin Duvalsaint[5], Jenny Legac[5], Moses R. Kamya[3,6], Grant Dorsey[5], Francesca Aweeka[1], Philip J. Rosenthal[5] & Rada M. Savic [2]✉

Intermittent preventive treatment (IPT) with dihydroartemisinin-piperaquine (DP) is highly protective against malaria in children, but is not standard in malaria-endemic countries. Optimal DP dosing regimens will maximize efficacy and reduce toxicity and resistance selection. We analyze piperaquine (PPQ) concentrations ($n = 4573$), malaria incidence data ($n = 326$), and *P. falciparum* drug resistance markers from a trial of children randomized to IPT with DP every 12 weeks ($n = 184$) or every 4 weeks ($n = 96$) from 2 to 24 months of age (NCT02163447). We use nonlinear mixed effects modeling to establish malaria protective PPQ levels and risk factors for suboptimal protection. Compared to DP every 12 weeks, DP every 4 weeks is associated with 95% protective efficacy (95% CI: 84–99%). A PPQ level of 15.4 ng/mL reduces the malaria hazard by 95%. Malnutrition reduces PPQ exposure. In simulations, we show that DP every 4 weeks is optimal across a range of transmission intensities, and age-based dosing improves malaria protection in young or malnourished children.

[1] Department of Clinical Pharmacy, University of California, San Francisco, San Francisco, CA, USA. [2] Department of Bioengineering and Therapeutic Sciences, University of California, San Francisco, San Francisco, CA, USA. [3] Infectious Diseases Research Collaboration, Kampala, Uganda. [4] Department of Medicine, Stanford University, Palo Alto, CA, USA. [5] Department of Medicine, University of California, San Francisco, San Francisco, CA, USA. [6] Department of Medicine, Makerere University, Kampala, Uganda. ✉email: rada.savic@ucsf.edu

In malaria-endemic regions young children bear the greatest burden of malaria, primarily due to *Plasmodium falciparum*, including severe malaria and death[1]. In Uganda, nearly 75% of infants in one study developed malaria before 1 year of age[2], and by 2 years of age, an average malaria incidence exceeding 6 episodes per year has been reported[3]. Prompt effective malaria treatment, long-lasting insecticidal bednets (LLINs), and indoor residual spraying of insecticides (IRS) have been the mainstays of malaria control for young children, accompanied by decreases in the global malaria burden[1]. However, reductions in malaria incidence and mortality have stalled, and new malaria control interventions are needed[1].

Intermittent preventive treatment (IPT), in which full antimalarial treatment courses are given at fixed intervals to prevent malaria, is utilized to reduce malaria incidence in vulnerable populations. Seasonal malaria chemoprevention, in which children receive monthly sulfadoxine-pyrimethamine (SP) and amodiaquine during malaria transmission months, has reduced malaria incidence and mortality in children in parts of the west Africa[4,5]. For children residing in most of Africa, where SP resistance is already widespread and malaria transmission is perennial, a highly effective WHO or country-approved IPT regimen is not available, and in Uganda IPT with SP was ineffective[3,6]. A novel IPT drug regimen is needed for African children.

The artemisinin-based combination therapy dihydroartemisinin-piperaquine (DP) is the leading candidate for IPT in young children in malaria-endemic regions with widespread SP resistance. Dihydroartemisinin rapidly clears parasites, while piperaquine (PPQ), the longer-acting aminoquinoline partner drug, eliminates residual parasites and provides sustained protection against new infections for approximately one month after treatment[7]. Clinical trials among children in Uganda showed that monthly IPT with DP was superior to placebo or to DP every 12 weeks in young children, with protective efficacies >95%[3,6,8]. When administered to school-age children, IPT with DP every 4 weeks lowered community level parasite prevalence[9]. Large studies have not identified significant toxicities associated with IPT with DP in children, even though plasma PPQ concentration has been positively associated with lengthening of the corrected QT interval (QTc)[10,11]. Despite the appeal of DP for IPT, the optimal dose and frequency, particularly in children less than 2 years of age, is not well established. Underdosing of DP for IPT could result in inadequate preventive efficacy and selection of antimalarial drug resistance, while overdosing could increase cost and the risk of toxicity.

In this work, to gain insights into optimal DP regimens for IPT in young Ugandan children, we leveraged data from a randomized controlled trial and used pharmacokinetic/pharmacodynamic (PK/PD) modeling to describe the pharmacokinetics (PK) of PPQ and characterize relationships between PPQ exposure and risks of malaria, toxicity, and selection for markers of aminoquinoline resistance. Furthermore, we conducted simulations to quantify how optimized DP dosing regimens between 2 and 24 months of age would decrease malaria incidence, risk of QTc prolongation, and selection for *P. falciparum* markers of decreased antimalarial drug sensitivity.

## Results

**Study population and raw data.** Among the 280 children who received at least 1 course of DP, 243 (87%) completed follow up through 36 months of age (Fig. 1 and 2). Participant characteristics were similar between the two IPT arms, with the exception that none of the children who received DP every 4 weeks were born to mothers who received IPT with SP during pregnancy as per the trial protocol (Table 1). All participants had at least one

PPQ concentration determined (median number [range] per participant: 31 [16–33] for intensive PK sampling; 12 [1–20] for sparse PK sampling). There were 4573 PPQ concentrations quantified; 578 (12.6%) were below the lower limit of quantification (BLQ). The distribution of PPQ concentrations is shown in Fig. 3.

Malaria incidence was significantly lower in children receiving IPT from 8 to 104 weeks of age every 4 weeks compared to every 12 weeks (0.017 vs 0.322 malaria episodes/per person-year, incident rate ratio (IRR): 0.05 [95% CI: 0.012–0.16]). The cumulative risk of malaria after receipt of DP was 8% (6.7–9.3%) through 84 days after dosing for DP every 12 weeks and 0.1% (0.0–0.30%) through 28 days after dosing for DP every 4 weeks. In the DP every 12 weeks arm, cumulative risk of malaria varied by transmission period, with high transmission periods in 2016 and 2017 associated with significantly higher risks of malaria compared to low transmission periods (hazard ratio [HR] 2016: 5.85, 95% CI 3.79–8.95, p < 0.001; HR 2017 7.16, 95% CI 4.20–12.2, p < 0.001; Fig. 4B); there was no transmission period effect in the DP every 4 weeks arm (Fig. 4C). The median PPQ concentration at the time of incident malaria was 2.0 ng/mL (2.5–97.5%: BLQ–12.9 ng/mL), which was significantly lower than the median PPQ concentration 28 days after the last DP dose (4.9 ng/mL, 2.5–97.5: 0.88–18.4 ng/mL), but higher than the 56 and 84 days post DP concentrations in the every 12 weeks arm (Fig. 5A).

**PK model building.** For the PPQ population PK model, a 3-compartment distribution model (ΔOFV −200.1 compared to 2 compartments) with 2 transit absorption compartments provided the best description of the PK data (Supplementary Fig. 1A, B). A log-linear relationship best described the relationship between venous and capillary concentrations, and the relationship did not vary by age during the study period (Supplementary Fig. 2). The BLQ/2 method (M6 method) and estimation of the likelihood of BLQ (M3 method) provided similar parameter estimates, and so the BLQ/2 method was used to handle BLQ data, with BLQ PPQ measurements assigned to 0.25 ng/mL.

Several PK covariates were identified. PPQ is primarily metabolized by cytochrome P450 3A4[7], and a maturation function using postmenstrual age (PMA) to represent liver enzyme maturation during early childhood, significantly modified PPQ clearance as shown in Eq. 1, where $\theta_{CL}$ is the population clearance, $\theta_{EC50}$ is the PMA when maturation reaches 50%, and $\eta$ is the between-subject random term (ΔOFV-331; Table 2).

$$CL = \theta_{CL} * \left(\frac{WEIGHT}{8.6\,kg}\right)^{0.75} * \left(\frac{PMA}{PMA + \theta_{EC50}}\right) * e^{\eta} \quad (1)$$

A higher degree of malnutrition, as measured by either lower weight for age *z*-score (WAZ), height for age *z*-score (HAZ), or weight for height *z*-score (WHZ), all reduced PPQ bioavailability (ΔOFV WAZ −22.1, ΔOFV HAZ −5.93, ΔOFV WHZ −6.47). Malnutrition, as measured by WAZ provided the greatest statistical significance, the greatest covariate effect size (for each SD decrease in *z*-score there was a decreased bioavailability by 11.3% for WAZ vs 4.7% for HAZ or 4.4% for WHZ), and the best model fit by visual predictive check. Therefore, WAZ was selected as the covariate to represent malnutrition in the final PK model. In the final model, each standard deviation decrease in WAZ, decreased the relative oral bioavailability by 11.3%. In addition, when DP was self-administered there was lower PPQ oral bioavailability (ΔOFV −94.8). All three daily doses of DP were directly observed for participants in the intensive PK substudy, otherwise, the first DP dose was administered in the clinic, and the remaining 2 were given to the guardian to be administered at

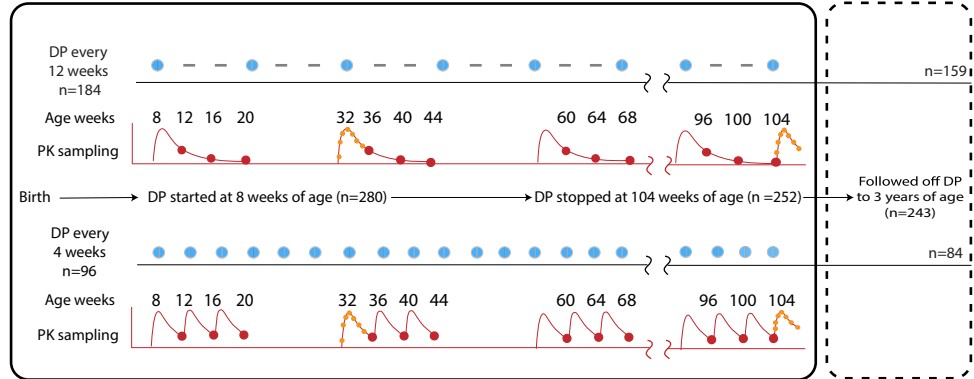

**Fig. 1 Summary of trial interventions and sampling for piperaquine concentrations.** Blue dots indicate dihydroartemisinin-piperaquine (DP) courses dispensed, and dashes indicate episodes when a placebo was given. The red dots indicate sparse pharmacokinetic (PK) sampling for piperaquine concentrations (all participants contributed data) and red lines indicate typical pharmacokinetic profiles. Intensive sampling, in orange dots, occurred at 32 and 104 weeks (22 individuals in every 12 weeks and 10 in the every 4 weeks arm).

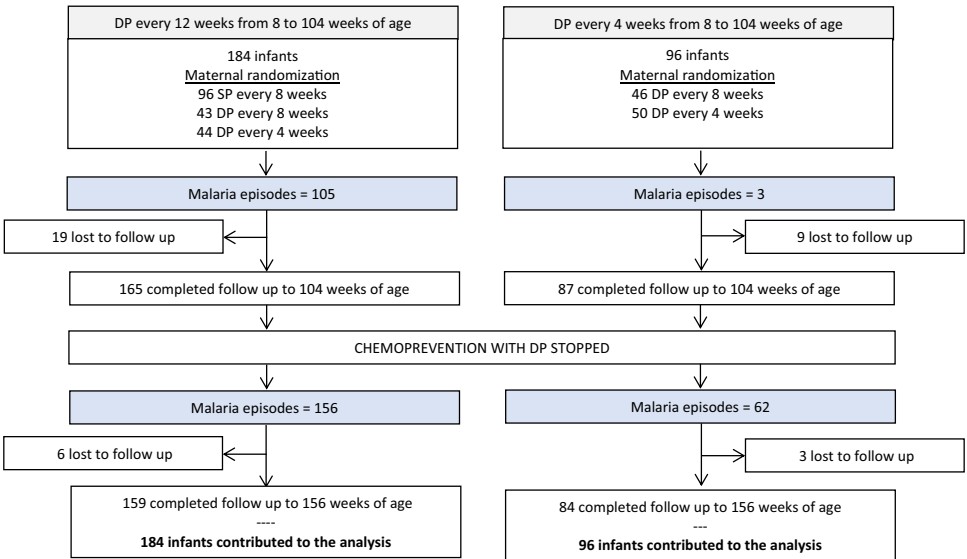

**Fig. 2 Study participant follow-up and malaria outcomes in the clinical trial.** DP indicates dihydroartemisinin-piperaquine.

home. Compared to the intensive DP dosing episodes, oral bioavailability was 60% lower if the DP doses were taken at home (Table 2 and Supplementary Fig. 3), indicating that some DP doses were missed without directly observed therapy. Oral bioavailability is described in Eq. 2, where $\theta_F$ is the population bioavailability, which was assumed to be 100%, $\theta_{WAZ}$ is the parameter for time-varying WAZ, which was centered on the median value, $\theta_{Self-administered\,DP}$ is the covariate representing self-administered IPT dosing episodes, and BOV is between occasion variability.

$$F = \theta_F * (1 + \theta_{WAZ}*(WAZ - (-0.5))) * \theta_{Self\,administered\,DP} * e^{BOV}$$
(2)

A mixture model was tested to evaluate for participant populations with either high or low apparent bioavailability, but distinct populations could not be identified. The maternal chemoprevention regimen was not associated with PK exposure.

**PK/PD model for incident malaria.** Parametric survival analysis, adjusted for repeated events, was used to predict the hazard of incident malaria from 2 to 36 months of age. An exponential distribution provided the best baseline model, and significant covariates for malaria hazard included: a categorical transmission

period adjustment to account for higher malaria transmission calendar months, time-varying plasma PPQ concentration, and maternal socioeconomic status (SES). Transmission period and time-varying PPQ concentration were included in the final model as shown in Eq. 3, where $\theta_{Baseline}$ is the baseline hazard, $\theta_{Transmission\,period}$ is an adjustment for observed annual transmission increases in malaria transmission, $\theta_{EC50}$ is the EC50 for PPQ, $\gamma$ is the Hill coefficient, [PPQ] is PPQ concentration, and $\eta$ is the between-subject random term.

$$\text{Hazard of incident malaria} = \theta_{Baseline} * \theta_{Transmission\,period} * \left(1 - \frac{[PPQ]^\gamma}{\theta_{EC50}^\gamma + [PPQ]^\gamma}\right) * e^\eta$$
(3)

Three increases in malaria incidence corresponding to the annual high transmission periods in Tororo occurred (Fig. 4A), and we adjusted the baseline hazard for incident malaria during these periods (ΔOFV −431). Higher PPQ concentrations reduced the risk of malaria with a maximum effect of 100% reduction in hazard (ΔOFV −181). The half-maximal protective concentration (EC50) of PPQ was 6.0 ng/mL (Table 2). A PPQ concentration of 15.4 ng/mL reduced the hazard of incident malaria by 95% (Fig. 5B), and was used as the target PPQ concentration for DP regimen simulations. Maternal SES, as defined by a propensity

**Table 1 Patient characteristics.**

**DP regimen (received from age 8 to 104 weeks)**

| Characteristic | DP every 12 weeks | DP every 4 weeks |
|---|---|---|
| Number randomized | 184 | 96 |
| Female sex, *n* (%) | 92 (50%) | 45 (47%) |
| Birth weight, median (2.5%, 97.5%) | 3000 (1932–3807) | 2965 (1694–3688) |
| Gestational age, median (2.5%, 97.5%) | 39.9 (33.2–41.4) | 39.0 (33.0–41.8) |
| Low birth weight (<2500 gr), *n* (%) | 21 (11.4%) | 14 (14.6%) |
| Preterm birth (<37 weeks), *n* (%) | 14 (7.6%) | 12 (12.5%) |
| Maternal IPTp regimen, *n* (%) | | |
| SP every 8 weeks | 97 (53%) | – |
| DP every 8 weeks | 43 (23%) | 46 (48%) |
| DP every 4 weeks | 44 (24%) | 50 (52%) |
| Weight for age *z*-score at age 8 weeks, median (2.5%, 97.5%) | −0.22 (−2.92–1.36) | −0.31 (−3.75–1.21) |
| Height for age *z*-score at age 8 weeks, median (2.5%, 97.5%) | 0.03 (−3.27–2.06) | −0.39 (−4.38–1.18) |
| Sparse sampling—PPQ concentrations (280 children) | | |
| Routine[a] Venous, *n* (% eligible) | 378 (97%) | 166 (91%) |
| Routine[a] Capillary, *n* (% eligible) | 1890 (99%) | 945 (99%) |
| Non-routine[b] PPQ concentrations, *n* | 200 | 25 |
| Intensive sampling—PPQ concentrations (32 children) | | |
| Venous, *n* | 403 | 180 |
| Capillary, *n* | 273 | 113 |
| *Plasmodium falciparum antimalarial resistance genotypes from first episode of parasitemia after DP[c]* | | |
| Episodes of parasitemia through 112 weeks of age | 135 | 17 |
| *pfmdr*1 86Y (%) | | |
| Successful genotypes, (%) | 122 (90%) | 12 (71%) |
| Mutant infections[c], (%) | 9 (7.4%) | 1 (8.3%) |
| *pfmdr*1 184F (%) | | |
| Successful genotypes, (%) | 130 (96%) | 12 (71%) |
| Mutant infections[c], (%) | 79 (60.8%) | 6 (50%) |
| *pfmdr1* 1246Y (%) | | |
| Successful genotypes, (%) | 121 (90%) | 10 (59%) |
| Mutant infections[c], (%) | 26 (21%) | 1 (10%) |
| *pfcrt* 76T (%) | | |
| Successful genotypes, (%) | 122 (90%) | 9 (53%) |
| Mutant infections[c], (%) | 47 (39%) | 3 (33%) |

[a]Routine indicates PPQ concentrations taken at pre-specified study visits.
[b]Non-routine PPQ concentrations were taken at non-specified study visits (i.e., at the time of parasitemia).
[c]Mutant parasites included polyclonal infections with wild-type and mutant and pure mutant infections, only the first infection detected after receiving a course of DP was considered for genotyping.

score summarizing property and income, was assigned a value between −1 and 3. In univariate analysis, we found that each 1 unit increase in maternal SES was associated with a 26.2% decreased risk of malaria (ΔOFV −7.21). However, when we incorporated SES into the full PK/PD model we encountered unacceptable model instability and confidence intervals could not be reliably acquired by bootstrap, so maternal SES was not included in the final model. A semi-mechanistic model was explored which incorporated parasite replication rates extrapolated from experimental infection studies in malaria naïve adult populations[12,13], which would enable us to predict PPQ concentrations at the time of liver emergence. We found that in our study population, the semi-mechanistic model did not predict the data well, and the empirical model was used as the final model. Sex, IPT arm, maternal IPT regimen, WAZ, WHZ, and HAZ were not associated with the hazard of incident malaria.

**PK-QTc model**. To assess relationships between PPQ concentration and risk of QT interval by Bazett's correction (QTcB) prolongation, a PK-QTc model was developed based on data from the intensive PK substudy with paired ECGs from 32 participants at 32 and 104 weeks of age. As previously reported, the median QTcB pre-drug was 413 msec (range: 347–472), and post-drug was 424 msec (range: 388–482). Increasing PPQ concentration increased the QTcB as described in the following linear

equation: QTcB = modeled baseline QTcB + [PQ] × 0.046/1000. Each 100 ng/mL increase in PPQ concentration was associated with a 4.6 msec increase in the QTcB (Supplementary Table 2 and Supplementary Fig. 4).

**PK/PD resistance model**. We assessed relationships between PPQ concentration and probability of detecting infections with *P. falciparum* containing mutations associated with decreased aminoquinoline sensitivity, including in *pfmdr1*, the gene that encodes multidrug resistance protein 1 (*PF3D7_0523000*), and in *pfcrt*, the gene that encodes the chloroquine resistance transporter (*PF3D7_0709000*). The following polymorphisms were evaluated: *pfmdr1* N86Y, *pfmdr1* Y184F, *pfmdr1* D1246Y and *pfcrt* K76T[14]. Genotype data were available from 142 episodes of parasitemia (88% of eligible episodes) from 8 to 112 weeks of age (Table 1). There were no significant differences in the prevalence of mutant parasites between every 12-week and every 4-week IPT arms. Time-varying PPQ concentration was not significantly associated with the probability of detecting a mutant parasite when parasitemia was detected.

**Simulations**. For each regimen, 1000 simulations of the PK model and 10,000 simulations of the parametric survival model were conducted using longitudinal demographic data from 856 Ugandan children (280 children who contributed data to this

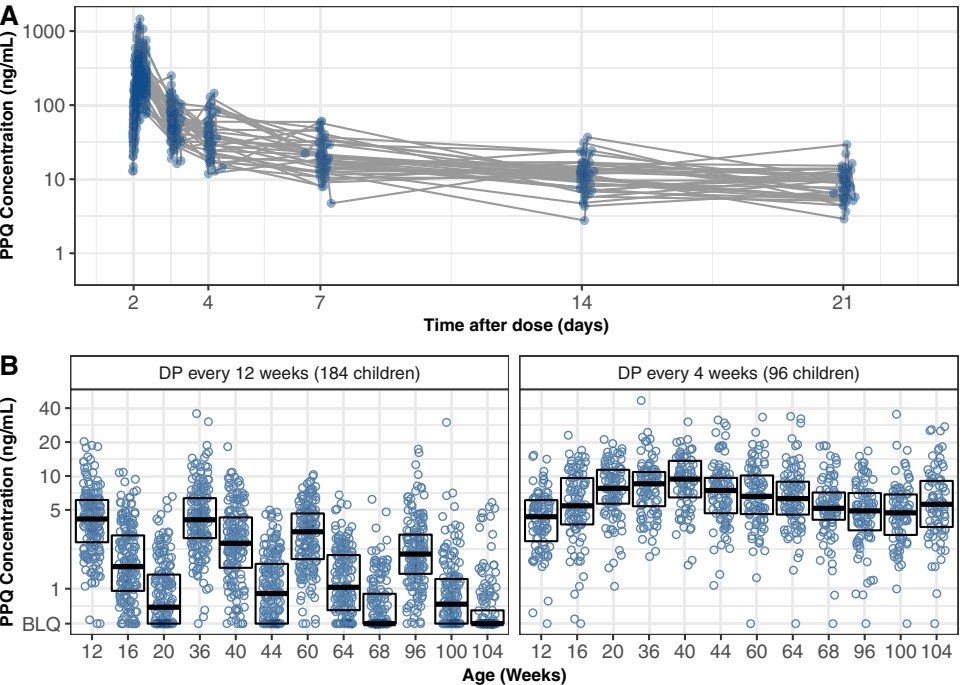

**Fig. 3 Raw pharmacokinetic data. A** Piperaquine (PPQ) concentration from intensive sampling after the third daily dihydroartemisinin-piperaquine (DP) dose (day 2) for 32 children at 32 and 104 weeks of age. **B** PPQ concentrations from sparse sampling obtained from 280 children at 28-days intervals. Boxes indicate PPQ levels for 25% (minima), 50% (center), and 75% (maxima) of the population.

analysis and 576 children from 6 months to 2 years of age from two prior study cohorts from the same region[3,6]. Time above protective PPQ concentrations and clinical malaria incidence were calculated. Every 4-weeks regimens were predicted to be superior to every 8-weeks regimens by predicted percent time above protective PPQ concentrations (Table 3) and predicted incidence per person-year on IPT (Supplementary Fig. 5). Malnourished children with a WAZ ≤ −2 at the time of DP dosing, were predicted to have a lower percentage of time above protective PPQ concentrations and a resultant increased risk of clinical malaria compared to children with a WAZ > −2 (Table 3 and Fig. 6). In addition, trough PPQ concentrations decreased as children aged, with the lowest trough concentrations predicted after 22 months of age. Age-based dosing was predicted to increase the proportion of trough concentrations above 15.4 ng/mL, in particular, for children greater than 1 year of age (Fig. 6A). The age-based regimen was also predicted to reduce the incidence of clinical malaria comparing malnourished and nourished children across transmission intensities (Fig. 6B). Finally, maximum PPQ concentrations in children from 2 to 24 months of age were evaluated from simulations of the 280-child study population. The median maximum PPQ concentration during chemoprevention was predicted to be higher for both malnourished (141 ng/mL greater) and nourished children (77 ng/mL greater) with an age-based DP regimen compared to the currently approved WHO 2015 regimen (Fig. 6C). As peak PPQ concentrations for the higher dosed regimens (WHO 2015 and age-based) were higher than those used to develop the PK-QTc model, extrapolating the PK-QTc model for these populations must be interpreted with caution (Supplementary Fig. 6).

## Discussion

We report a PK/PD analysis based on the largest longitudinal clinical trial of IPT with DP in young children, including longitudinal pharmacokinetic, incident malaria, electrocardiographic, and drug sensitivity data. For children <2 years of age, we identified a malaria preventive PPQ concentration of 15.4 ng/mL. Children 1–2 years of age had the highest risk of malaria in the cohort, and had lower PPQ concentrations compared to children <1 year of age. Malnourished children were at particular risk of low PPQ exposure, with children with WAZ ≤ −2 predicted to have an increased incidence of malaria compared to better nourished children. Adherence to the three-day DP regimen was lower than expected, but despite low adherence DP given every 4-weeks had a protective efficacy of 95% compared to DP every 12-weeks. Based on these data, we predicted that a 4-week dosing interval with age-based, instead of weight-based, dosing would provide maximal protection and reduce disparities in PPQ exposure due to age and nutritional status.

PPQ concentrations predict malaria protective efficacy in a variety of high-risk populations receiving chemoprevention with DP, and identifying protective concentrations is essential to our effort to expand the role of DP to IPT. The 15.4 ng/mL protective PPQ target derived from this study is similar to protective levels against symptomatic malaria identified in children <5 years of age receiving DP for seasonal malaria chemoprevention in Burkina Faso (12.9–17.5 ng/mL)[13] and Thai adults (20 ng/mL)[12]. Among highly malaria-exposed Ugandan pregnant women, a level of 10.3 ng/mL was associated with the prevention of both symptomatic and asymptomatic parasitemia[15]. Despite differences in age, immune status, and region, with potential differences in drug resistance patterns or population genetics, these PPQ targets are remarkably similar. We found that median PPQ trough concentrations in young children were at or below 10–20 ng/mL with DP dosed every 4 weeks. As a result, we would expect that even a short delay in DP dosing frequency beyond 4-weeks would result in a loss in protective efficacy. Similar challenges have been observed with SMC implementation with SP with amodiaquine, where malaria incidence reached levels similar to no chemoprevention by 29–35 days after a round of SMC [16].

Malnourished children were at the highest risk of suboptimal PPQ exposure and were predicted to suffer the highest incidence

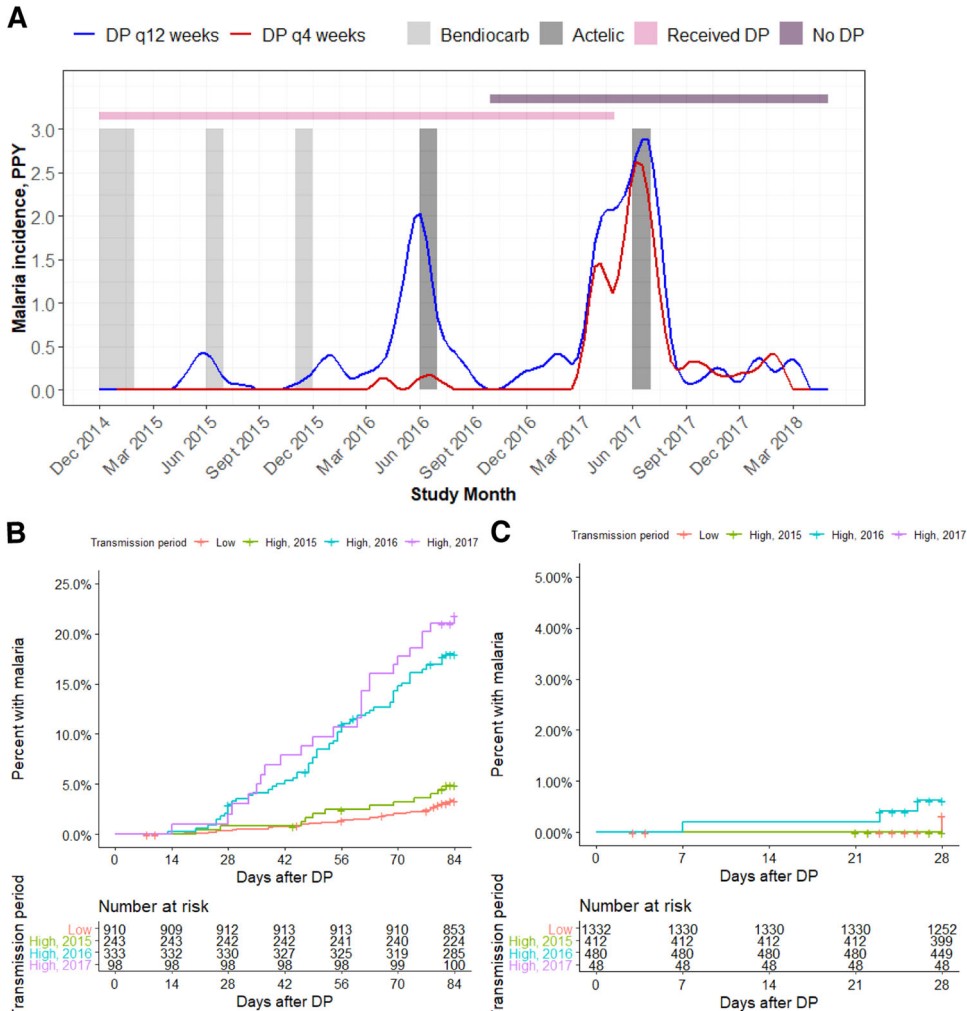

**Fig. 4 Malaria outcomes data. A** Malaria incidence per person year, stratified by dihydroartemisinin-piperaquine (DP) regimen. The lines indicate malaria incidence for the DP every 12 weeks ($n = 184$) (blue) and every 4 weeks ($n = 96$) (red) arms. Gray bands indicate periods of indoor residual spraying, and horizontal bars indicate times when participants received DP (pink) or were monitored off DP (purple). As study enrollment occurred over several months, there was a period where part of the cohort was on DP and others had completed the intervention (overlap between pink and purple bars). Time to malaria after receiving DP in the every 12-week arm **B** and the every 4-week arm **C** stratified by malaria transmission period. The low transmission periods are combined and the high transmission periods are shown separately.

of malaria when receiving IPT with DP. Lower PPQ exposure was related to lower oral bioavailability with malnutrition. Lower oral bioavailability has been linked to malnutrition, as defined by low WAZ, among children <5 years of age for lumefantrine and SP[17,18]. A variety of biomarkers of acute malnutrition including mid-upper arm circumference (not available for this study), WHZ, and WAZ have been linked to lower antimalarial drug exposure for malaria treatment. More research is needed to elucidate the pathophysiologic mechanism for these findings[17,19,20]. Several physiologic changes due to malnutrition have been implicated, including increased intestinal inflammation leading to reduced drug absorption and low plasma albumin concentrations resulting in altered protein binding[21]. We found that even small reductions in WAZ lead to lower PPQ bioavailability. Although weight-based dosing strategies can incorporate dose increases to compensate for liver maturation in infancy, low weight malnourished children can fall into weight-bands designed for younger children, leading to unintentional underdosing[22]. This can be further exacerbated when children, as is the case of this study, between 1 and 2 years of age, were already underdosed.

We propose an age-based dosing algorithm that would both increase PPQ exposure in older children, compared to the dosing regimen used in the parent trial, and reduce the impact of malnutrition on PPQ exposure. Notably, we found that easily implementable revised weight or age-based DP dosing strategies did not fully eliminate the impact of malnutrition on PPQ exposure (Table 3). Thus, correcting underlying malnutrition would be needed to fully equalize PPQ exposure between malnourished and nourished children. With age-based dosing, 35% of children would have received higher daily doses of DP, while 2.5% would have received lower doses, compared to the WHO 2015 weight-based malaria treatment guidelines for DP.

By identifying malaria protective PPQ concentrations for young children, we predicted that if malaria transmission had been higher, such as would be expected in similar regions not receiving IRS, the incidence of malaria on IPT would have been higher. In the case of high malaria transmission, in addition to age and nutritional status, adherence would drive DP protective efficacy. Adherence is a well-known barrier to effective IPT, and low adherence has been observed with multiday malaria treatment and IPT regimens[23–25]. The parent clinical study was no

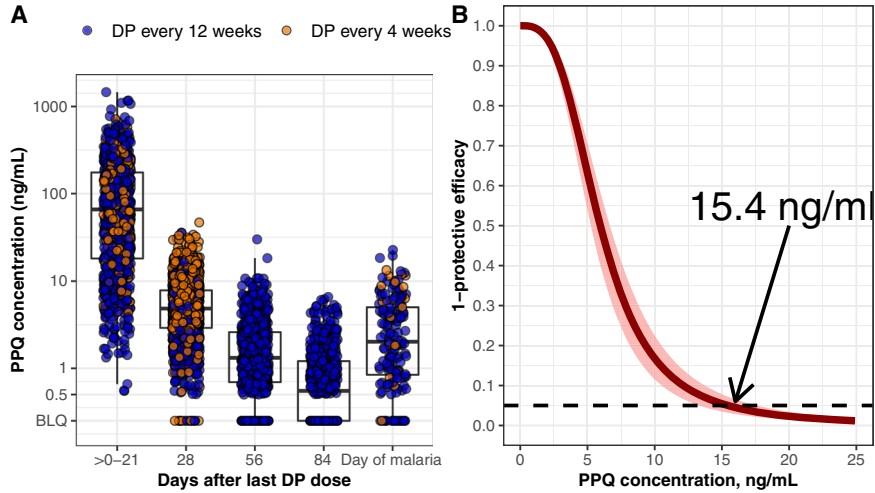

**Fig. 5 Relationship between piperaquine (PPQ) concentration and protective efficacy. A** Measured PPQ concentrations by days after dihydroartemisinin-piperaquine (DP) and at time of malaria diagnosis ($n = 280$ children). Points indicate observed data, boxes indicate PPQ levels for 25% (minima), 50% (center), and 75% (maxima) of the population, and vertical bars represent PPQ levels for 95% of the population. **B** Relationship between PPQ concentration and protective efficacy. The solid line indicates the median estimate from the pharmacokinetic/pharmacodynamic model, and the shaded areas the 95% confidence interval. The indicated concentration (15.4 ng/mL) reduced the hazard of incident malaria by 95%.

| Table 2 Pharmacokinetic and pharmacodynamic parameters. | | |
| --- | --- | --- |
| **Parameter** | **Value (%RSE, 95% CI)** | **Interindividual variability, % (%RSE, 95% CI)** |
| *Pharmacokinetic parameters* | | |
| N (subjects/PK observations) | 280/4573 | |
| Clearance (L/d)[a] | 867 (10.2%, 727–1064) | 27.1 (15.1%, 22.0–31.0) |
| $\Theta_{\text{Post menstrual age EC50}}$[a] (weeks) | 96 (15.4%, 73–130) | – |
| Volume of central compartment (L) | 592 (10.1%, 484–717) | 32.8 (56.3%, 11.8–48.5) |
| Intercompartmental clearance 1 (L/d) | 511 (12.4%, 401–645) | – |
| Volume of peripheral compartment 1 (L) | 7240 (8.2%, 6210–8440) | – |
| Intercompartmental clearance 2 (L/d) | 671 (16.8%, 461–934) | – |
| Volume of peripheral compartment 2 (L) | 1060 (18.2%, 731–1510) | – |
| Absorption transit time (d)[b] | 0.045 (9.1%, 0.034–0.048) | 43.2 (42.2%, 22.4–60.1) |
| $\Theta_{\text{Capillary to venous conversion}}$[c] | 0.922 | – |
| Proportional error | 44.6% (1.7%, 43.0%–46.0%) | – |
| Relative bioavailability (F)[b] | 1 | |
| $\Theta_{\text{Weight for age } z\text{-score}}$[d] | 0.113 (18.9%, 0.061–0.137) | – |
| $\Theta_{\text{Self-administered DP}}$[d] | 0.397 (7.8%, 0.344–0.465) | – |
| $\Theta_{\text{Between occasion variability}}$ | 66.9% (5.9%, 62.3%–71.3%) | |
| *Pharmacodynamic parameters* | | |
| N (subjects/observations) | 280/326 | |
| Baseline hazard/1000 (per day)[e] | 0.402 (17.2%, 0.268–0.536) | 69.6 (66.1, 43.3–146) |
| Season adjustment | | |
| $\Theta_{\text{Transmission period 2015}}$ | 1.29 (33.9%, 0.65–2.40) | – |
| $\Theta_{\text{Transmission period 2016}}$ | 5.20 (16.1%, 3.92–7.09) | |
| $\Theta_{\text{Transmission period 2017}}$ | 7.83 (16.7%, 5.75–10.96) | |
| $\Theta_{\text{PPQ EC50}}$ (ng/mL) | 6.00 (14.4%, 4.25–7.58) | – |
| $\Theta_{\text{PPQ } \gamma}$ | 3.13 (24.3%, 1.96–4.89) | |

Confidence intervals obtained by bootstrap ($n = 1000$).

[a]$CL = \Theta_{CL} * \left(\frac{\text{Weight}}{8.6\,\text{kg}}\right)^{0.75} * \left(\frac{\text{Post menstrual age}}{\theta\,\text{Post menstrual age EC50+Post menstrual age}}\right)$.

[b]Pre-specified absorption compartments.

[c]$\ln([PPQ]_{\text{capillary}}) = \Theta_{\text{slope}} * \ln([PPQ]_{\text{venous}})$.

[d]$F = 1* \Theta_{\text{Self-administered DP}} *(1 + \Theta_{\text{WAZ}}*(\text{WAZ}-(-0.5)))*e^{\Theta_{\text{ Between occasion variability}}}$, WAZ weight for age $z$-score.

[e]Survival function: $\Theta_{\text{Baseline}}/1000* \Theta_{\text{Transmission period}} * \frac{[PPQ]^{\gamma}}{\theta EC50^{\gamma}+[PPQ]^{\gamma}}$.

exception. Although >95% adherence was self-reported among study participants, on average PPQ absorption was 60% lower when only 1 dose, compared to all 3 doses of DP was directly observed. Although PPQ absorption is heterogeneous between individuals and dosing occasions[13] and can be increased

modestly by food[26,27], the magnitude of the association with self-administration is most consistent with a low adherence effect.

Despite lower than anticipated PPQ exposure, malaria incidence during the trial was lower than expected when compared to

**Table 3 Time above protective PPQ concentrations by adherence status and weight for age z-score at the time of most recent DP course.**

| Regimen | % Time above protective PPQ levels, (2.5–97.5% of population) | | | | | | | | |
|---|---|---|---|---|---|---|---|---|---|
| | Full adherence | | | 2/3 adherence | | | 1/3 adherence | | |
| | WAZ ≤ −2 | WAZ > −2 | Percentage of children with >80% time above 15.4 ng/mL | WAZ ≤ −2 | WAZ > −2 | Percentage of children with >80% time above 15.4 ng/mL | WAZ ≤ −2 | WAZ > −2 | Percentage of children with >80% time above 15.4 ng/mL |
| **Every 4-week DP** | | | | | | | | | |
| Clinical trial protocol | 67.3 (14.2–100) | 92.4 (19.2–100) | 56.6 (54.4–58.8) | 36.7 (8.8–100) | 60.1 (11.9–100) | 35.8 (32.4–39.5) | 16.4 (3.9–98.7) | 23.9 (5.4–100) | 12.0 (10.2–14.3) |
| WHO 2015 | 84.6 (18.0–100) | 100 (29.8–100) | 74.5 (72.3–76.5) | 50.1 (11.1–100) | 86.0 (17.5–100) | 51.1 (48.9–53.3) | 20.8 (5.0–100) | 37.7 (8.0–100) | 21.2 (19.5–23.0) |
| Proposed age-based | 100 (28.9–100) | 100 (35.9–100) | 81.3 (79.4–83.1) | 81.9 (17.8–100) | 94.4 (20.6–100) | 59.5 (57.3–61.8) | 35.0 (7.9–100) | 45.4 (9.4–100) | 26.2 (24.4–28.2) |
| **Every 8-week DP** | | | | | | | | | |
| Clinical trial protocol | 19.7 (6.0–78.5) | 28.4 (7.8–96.3) | 6.3 (5.3–7.4) | 11.5 (3.8–54.8) | 15.9 (5.0–73.3) | 1.6 (1.1–2.1) | 6.2 (1.8–25.7) | 7.8 (2.4–39.4) | 0.1 (<0.01–0.3) |
| WHO 2015 | 26.3 (7.4–91.0) | 45.9 (11.4–100) | 15.4 (13.8–17.1) | 14.9 (4.8–69.3) | 26.5 (7.2–94.3) | 5.3 (4.4–6.3) | 7.7 (2.3–36.0) | 12.0 (3.6–62.0) | 0.7 (0.4–1.0) |
| Proposed age-based | 42.0 (11.2–100) | 50.9 (12.8–100) | 18.9 (17.0–20.7) | 24.5 (7.2–86.9) | 29.4 (8.1–95.7) | 6.5 (5.4–7.6) | 11.6 (3.5–55.6) | 13.4 (4.0–65.4) | 0.9 (0.5–1.2) |

WAZ weight-for-age z-score.
Clinical trial protocol: <6 kg: DHA/PPQ 10/80 mg daily × 3 days, 6–<11 kg: DHA/PPQ 20/160 mg daily × 3 days, 11–<15 kg: DHA/PPQ 30/240 mg daily × 3 days, 15–<20 kg: DHA/PPQ 40/320 mg daily × 3 days.
World Health Organization (WHO) 2015: <8 kg: DHA/PPQ 20/160 mg daily × 3 days, 8–<11 kg: DHA/PPQ 30/240 mg daily × 3 days, 11–<17 kg: DHA/PPQ 40/320 mg daily × 3 days, 17–<25 kg: DHA/PPQ 50/480 mg daily × 3 days.
Proposed age-based: 2–6 months of age: DHA/PPQ 20/160 mg daily × 3 days, 6–18 months of age: DHA/PPQ 30/240 mg daily × 3 days, 18–24 months of age: DHA/PPQ 40/320 mg daily × 3 days.

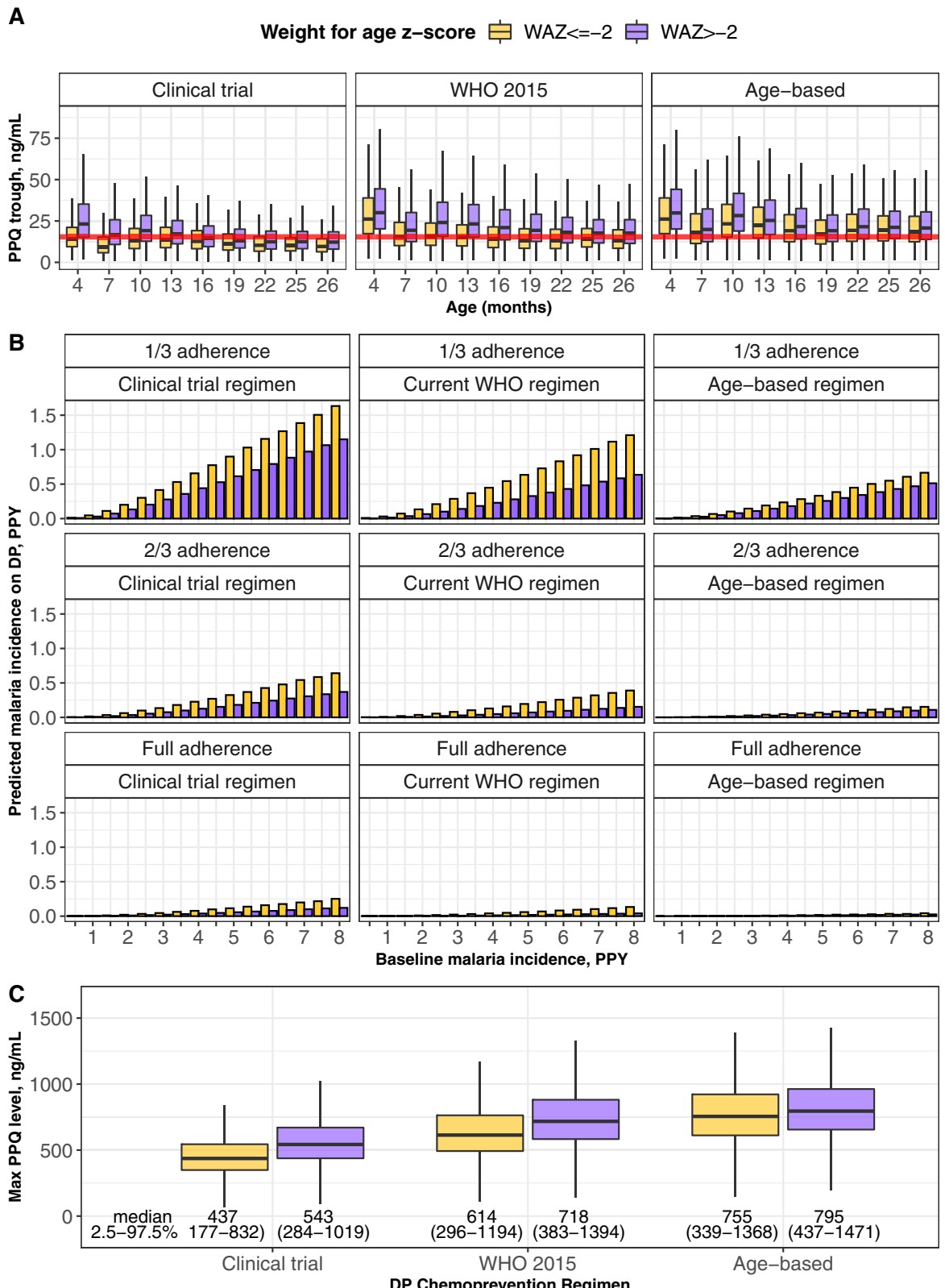

other studies of DP as IPT in young children, almost certainly due to recent malaria control efforts in the region[3,4]. Incident malaria was clustered during three short periods of high transmission, each timed late after a round of government-implemented IRS (Fig. 4A). Furthermore, IPT began at 2 months of age, when infants may still be protected against malaria from the transfer of maternal humoral immunity, low body surface area, and

increased use of malaria control measures such as LLINs[28,29]. To explore how DP regimens would perform in a variety of malaria transmission intensities and adherence patterns, we conducted simulations in a variety of adherence and transmission scenarios. In all of these scenarios, age-based dosing was predicted to provide the greatest malaria protective efficacy (Fig. 6), and we predicted that if the underlying malaria transmission intensity

**Fig. 6 Simulation results stratified by weight for age z-score (WAZ). A** Predicted piperaquine (PPQ) trough concentrations for simulated dihydroartemisinin-piperaquine (DP) regimens, stratified by nutritional status. Data are derived from 1000 simulations of 856 children <2 years of age. The red line indicates the 15.4 ng/mL PPQ target. Points indicate observed data, boxes indicate PPQ levels for 25% (minima), 50% (center), and 75% (maxima) of the population, and vertical bars represent PPQ levels for 95% of the population. **B** Predicted malaria incidence by DP regimen with increasing baseline malaria transmission, stratified by nutritional status and adherence level (1/3 adherence indicates bioavailability observed for non-direct observed therapy in the study, 2/3 adherence indicates a bioavailability midpoint between the directly and non-directly observed population, and full adherence indicates the bioavailability observed in the directly observed therapy group). Data are derived from 10,000 simulations of 856 children <2 years of age. **C** Predicted peak PPQ concentrations by DP regimen, assuming full adherence. Data are derived from 1000 simulations of 280 children <2 years of age. Points indicate observed data, boxes indicate PPQ levels for 25% (minima), 50% (center), and 75% (maxima) of the population, and vertical bars represent PPQ levels for 95% of the population. In text is the median and 2.5–97.5% range of predicted population values for peak PPQ concentrations during chemoprevention. Age-based dosing indicates daily PPQ doses as follows: <6 months = 160 mg; 6–<18 mo = 240 mg; 18–26 mo = 320 mg.

had been higher, an optimized DP regimen with full adherence would have been essential to achieve full protective efficacy with DP. An age-based DP regimen could have additional operational benefits for IPT in young children, where weighing a child increases the burden of community-based IPT implementation. As data from children >2 years of age were not available for PK model development and age was found to impact PPQ clearance up through 2 years of age, we could not use our model to predict how PPQ parameters would be altered for older aged children and we did not conduct simulations in children older than 2 years of age. Further study of optimal DP dosing is needed for malaria chemoprevention in older age groups.

We expected sub-protective PPQ concentrations to increase the risk of detection of more drug-resistant parasites when infections occur, as has been observed for pregnant women in Uganda[30]. However, we found no significant associations between PPQ concentrations and selection for quinoline resistance markers *pfmdr1* 86Y and *pfcrt* 76T. This is likely a result of two factors. First, as both trial regimens contained DP, the likelihood of differential selection of resistant parasites was small. Second, as an enhanced selection of mutant parasites, rather than de novo mutation, is the likely form of selection for resistance[31], the low prevalence of mutant parasites circulating at the time of the study limited the likelihood of resistance selection[32]. Importantly, artemisinin resistance is emerging in east Africa, supported by the identification of parasites with mutations in the K13 gene known or suspected of mediating resistance in southeast Asia in Rwanda[33,34] and Uganda[35,36]. Resistance to PPQ in southeast Asia is associated with mutations in *pfcrt* and amplification of *plasmepsin* genes that have generally not been described in Africa[37]. Our study was conducted in an area without the emergence of these mutations[35,36], but in the event of resistance, we would expect that higher PPQ concentrations would be needed to prevent malaria, and that infection during sub-protective PPQ levels would select for resistance. This concern highlights the importance of minimizing breakthrough malaria infections during IPT, continued evaluation of IPT preventive efficacy, and continued surveillance for drug resistance, as is ongoing.

In terms of safety, we explored the relationship between PPQ concentrations and changes in the QTcB interval for a subset of 32 individuals with paired PK and ECG data. Similar to studies of adults and children who received DP, we found each 100 ng/mL PPQ associated with a 4.6 ms increase in QTcB[38–40]. Despite the PK-QTc relationship observed with PPQ, a corresponding increased risk of cardiac events has not been described[10]. Furthermore, the largest study of the PK-QTc relationship of DP found that at PPQ concentrations >420 ng/mL there was minimal additional PPQ associated QT prolongation[38]. This study used a lower dose of PPQ than is currently recommended; nearly 75% of PPQ concentrations in the PK-QTc analysis were <450. As a result, it is likely that the predicted risk of QTcB >500 msec (1.90% for malnourished and 2.44% for nourished children, Supplementary Fig. 6) is an overprediction of the risk of QTcB

prolongation, as we used the model to predict QTc changes at PPQ concentrations beyond those measured in the data.

There were several limitations to this study. First, a placebo IPT arm was not included in the clinical trial, and thus the baseline hazard of malaria was dependent on data from the DP every 12-week arm adjusted for PPQ exposure. This approach may have under-estimated the baseline hazard of malaria during the study. Second, exposure to dihydroartemisinin, the short-acting partner drug in DP, was not quantified. To incorporate the parasiticidal activity of dihydroartemisinin into our model, we assumed cumulative survival returned to 100% when DP was given, as dihydroartemisinin resistance had yet to be detected in southeastern Uganda at the time of the study[35,36]. Third, we predicted adherence was lower than was self-reported in the study, as we could not correlate PK profiles with specific DP adherence patterns without an objective adherence measure (e.g., identify PK exposures associated with days missed or partial doses taken). We recommend future IPT studies incorporate supplemental measures for drug adherence (e.g., pill counts). Fourth, we did not implement a mechanistic model to estimate PPQ concentrations at the time of parasite emergence from the liver, which occurs some days prior to symptom onset. We explored back-extrapolating the time of parasite emergence using parasite density at malaria diagnosis and historic parasite replication rates in adults[13], but found this mechanistic approach prevented us from predicting the clinical data. As all participants received the same three-day treatment courses, PPQ concentrations that prevent clinical malaria would correlate with PPQ levels at liver emergence, and as a result, we only simulated three-day treatment courses for our optimized regimens. Finally, we could not include maternal SES as a covariate on the baseline malaria hazard in the final PK/PD due to model instability. Fortunately, though SES status reduced the intraindividual variability in our exploratory models, it did not modify the key PK/PD relationships or baseline malaria hazard estimates during chemoprevention. Future studies should consider an externally validated SES measure.

Large studies of seasonal malaria chemoprevention with SP plus amodiaquine in west Africa have been associated with dramatic reductions in malaria incidence and mortality in children <5 years of age[41,42]. However, despite a high burden of malaria in countries like Uganda, IPT in children is not yet recommended in east Africa, where SP resistance is widespread and seasonal approaches are not appropriate. The results of the parent clinical trial and this large PK/PD analysis assessing the drug exposure-response relationship for PPQ and malaria protection, risk of QTcB prolongation, and drug resistance markers confirms that DP every 4-weeks in children <2 years of age is effective and safe, and can be further optimized by utilizing age-based dosing bands. An age-based DP dosing strategy could have additional operational benefits for IPT, by eliminating the need to weigh infants receiving DP. We also found PPQ exposure was lower in malnourished and children 1–2 years of age, and that an age-based dosing strategy would particularly benefit these children.

Although DP every 4-weeks is highly effective for IPT in Africa, we show that there are simple and easily implemented dose modifications that could improve protection.

## Methods

**Study population.** A randomized controlled trial provided data and samples for the analysis[8]. Neonates, born to mothers enrolled in a separate trial of IPT during pregnancy in Tororo, Uganda[43], were enrolled at birth from October, 2014 to May, 2015, and followed for 36 months[8]. Informed consent was provided by the parent or guardian for every participant. The study protocol was approved by the Makerere University School of Biomedical Sciences Research and Ethics Committee, the Ugandan National Council for Science and Technology, and the University of California, San Francisco Committee on Human Research. The clinical trial registration number is NCT02163447.

**Study design and randomization.** Children were randomized prior to birth and received DP every 12 weeks or every 4 weeks from 8 to 104 weeks of age (Fig. 1). Children born from mothers who received DP for IPT during pregnancy were randomized to either DP every 4 or 12 weeks, whereas children born from mothers who received SP were all randomized to IPT with DP every 12 weeks in order to maximize the power of the parent study to detect differences in malaria incidence in childhood resulting from the IPT regimen received during pregnancy. A matched placebo was administered on weeks when DP was not scheduled in every 12-week arm. DP was administered once daily for three consecutive days and dosed by weight-band as per manufacturer's guidelines at the time of protocol approval (Supplementary Table 1). The first daily DP dose was administered in the clinic, and the remaining two doses were provided to the parent/guardian to give at home. Routine visits occurred every 4 weeks for clinical assessment, blood smear, blood spots for filter paper, and either venous or capillary blood collection for plasma PPQ quantification. Parents/guardians were encouraged to bring their child to the study clinic for all illnesses. Malaria was diagnosed if a participant had a temperature >38 °C or a history of fever in the last 24 h and a positive thick blood smear. Uncomplicated malaria was treated with artemether-lumefantrine. Concurrent use of other medications with antimalarial activity or which could interact with DP (cytochrome 3A4 inducers or inhibitors) were avoided. Parents/guardians were asked about adherence to the home doses of DP from the prior month at each subsequent routine visit. All participants were provided a LLIN at enrollment and district-wide IRS was conducted with the carbamate bendiocarb (December, 2014–February, 2015; June–July, 2015; November–December, 2015) followed by pirimiphos-methyl (Actellic; June–July, 2016; June–July, 2017) [8].

**Pharmacokinetic sampling.** PPQ concentrations were quantified from plasma obtained from finger-prick at pre-specified timepoints 12, 16, 20, 36, 44, 60, 64, 68, 96, and 100 weeks of age and from venipuncture at 40 and 104 weeks of age. At each time point samples were obtained prior to the subsequent monthly course of DP or placebo. PPQ concentrations were additionally quantified when an episode of malaria or asymptomatic parasitemia was diagnosed between 8 and 112 weeks of age. A subset of 32 individuals was enrolled in an intensive PK substudy which occurred at 32 and 104 weeks of age. As previously reported[44], for these subjects, PK samples were obtained by venipuncture at 0.5, 1, 2, 3, 4, 6, 8, and 24 h after the third daily DP dose, and by finger-prick at 24 h and at 4, 7, 14, and 21 days post-DP. An electrocardiogram (ECG) was obtained prior to the first dose of DP at 32 and 104 weeks of age, and 2–3 h after the 3rd daily dose of DP for all 32 children in the intensive PK substudy.

PPQ concentrations were determined from plasma using high-performance liquid chromatography and tandem mass spectrometry (HPLC-MS). Plasma for PPQ concentration quantification was separated from whole blood at the time of collection, and frozen immediately for storage at −80 °C. Plasma was shipped on dry ice to the UCSF Drug Research Unit in San Francisco, CA. PPQ quantification was conducted by protein precipitation using 25 μL of plasma, followed by liquid chromatography and tandem mass spectrometry. Calibrated standards ranging from 0.5 to 50 ng/mL were used and quality control (QC) samples comprise 1.5, 20, 40 ng/mL for sparse samples and standards from 10 to 1000 ng/mL and QC samples comprised of 30, 200, and 800 were used for the intensive PK analysis[45]. A coefficient of variation <10% for quality control samples was used. Qualitative interpretation of chromatographs to further evaluate BLQ samples was conducted by two experienced mass spectrometrists. BLQ samples with no observed peak, based on review by the mass spectrometrists, or which were measured after a prior BLQ measurement without interval dosing of DP were excluded from the analysis. If a peak was observed in a BLQ sample, it was considered as BLQ.

***P. falciparum*** **detection and genotyping.** A blood spot was collected and stored on filter paper at all routine visits and if malaria was diagnosed at an unscheduled visit. DNA was extracted from dried blood spots using Chelex-100, and all microscopy negative samples were tested for the presence of *P. falciparum* DNA by loop-mediated isothermal amplification (LAMP) as part of the parent study[8]. For the first new episode of parasitemia detected after a round of DP, genotyping for *pfmdr1* N86Y, *pfmdr1* Y184F, *pfmdr1* D1246Y, and *pfcrt* K76T was conducted.

Parasite DNA was extracted from a dried blood spot using Chelex-100, the gene of interest amplified using nested polymerase chain reaction, and polymorphisms detected using a ligase detection reaction-fluorescent microsphere assay[46]. A mutant infection at each locus was defined as detection of a mutant genotype, with or without concurrent detection of a wild-type genotype for a polyclonal infection. A wild-type infection was defined as detection of only *pfmdr1* N86, *pfmdr1* Y184, *pfmdr1* D1246, or *pfcrt* K76 in the *P. falciparum* positive sample.

**Population PK model.** All analyses were conducted in NONMEM version 7.4 or R version 3.6.1. We first established a model for venous plasma PPQ concentrations, followed by the addition of capillary PPQ concentrations to develop a joint model. We investigated 2-, 3-, and 4- compartment PK models linked to a first-order absorption model with lag time or absorption described by pre-specified transit compartments. Individual parameters were assumed to be normally distributed, and proportional and additive errors were evaluated for quantification of residual variability. Linear and log-linear models with and without an intercept were explored for the relationship between capillary and venous plasma PPQ concentrations. Clearance and volume parameters were allometrically scaled for bodyweight a priori by normalizing the child's weight to the median weight of the study population (8.6 kg) and raising to the power of 0.75 for all clearance parameters and to the power of 1 for all volume PK parameters. Relationships between pharmacokinetic parameters and covariates (age, time-varying HAZ, time-varying WAZ, time-varying WHZ, sex, adherence to DP, maternal chemoprevention regimen [SP, DP every 8 weeks, DP every 4 weeks], maternal education, and maternal SES) were assessed by graphical inspection and formal stepwise covariate model building. Validated methods for incorporating BLQ PPQ concentrations including the M1-7 methods were explored[47]. Model building was guided by the likelihood ratio test to determine statistical significance, diagnostic plots, and internal model validation techniques, including visual predictive checks [48].

**Exposure-response and derivation of PPQ concentrations for malaria protection.** Cox proportional hazard models were used as an initial evaluation of the raw data for cumulative malaria hazard by treatment arm. A parametric survival model, adjusted for repeated events was developed as the final model to predict the primary outcome, incident malaria. An incident malaria episode was defined as fever and positive blood smear >14 days from a prior episode of malaria (to minimize the effect of artemether-lumefantrine treatment failure). Exponential, Weibull, and Gompertz distributions were tested as the survival baseline model prior to evaluating covariates. Covariate analysis included time-varying PPQ concentration as defined by model-derived individual PK parameters, high malaria transmission period (defined as 1st March to 31st August annually), age, sex, time-varying WAZ, time-varying HAZ, time-varying WHZ, maternal IPT regimen during pregnancy, and maternal SES. Covariate relationships for continuous covariates included linear and nonlinear relationships (e.g., exponential, power, and Emax). Model building was guided by the likelihood ratio test, diagnostic plots, and visual predictive checks.

The PPQ concentration associated with protection from malaria was defined as the median PPQ concentration predicted to provide a 95% reduction in hazard of malaria as compared to no drug. Monte Carlo simulations were conducted to predict malaria incidence and time above protective PPQ concentrations under varying malaria transmission intensities and under different DP dosing regimens based on weight-band and age. The simulated regimens are listed in Supplementary Table 1, and included the clinical trial regimen which was dosed by weight-band as described by the package insert; the WHO 2015 treatment guidelines regimen which recommended increased DP doses by weight-band for children <25 kg[49]; and a novel age-based dosing regimen where DP doses were increased at 6 and 18 months of age for all children, regardless of weight. An age-based regimen was chosen as prior studies had identified malnutrition as a risk factor for under-exposure for PPQ in some populations and SMC is dosed by age to assist with large-scale implementation efforts[50,51]. For each regimen and adherence pattern (1/3, 2/3, or full adherence), the final PK model was simulated 1000 times and the final parametric survival model for malaria hazard was simulated 10,000 times using the full monthly demographic data from a combined dataset of the 280 study participants from 8 weeks to 24 months of age which data contributed to this study and from 576 children from 6 to 24 months of age enrolled in a previously conducted set of clinical trials in Tororo, Uganda[3,6]. The percentage of time above the model-derived protective PPQ concentration per DP treatment course was estimated for each regimen from the final PK model. Simulations of the final parametric survival model were conducted using baseline hazards from 0.5 to 8 episodes per person-year. Malaria hazard was kept constant in the simulations.

**PK-QTc analysis.** Pre- and post-dose QT intervals were obtained from ECGs from the 32 children who received intensive PK analysis at 32 and 104 weeks of age as previously reported[44]. The corrected QTc by Bazett's formula ($QT/\sqrt{RR}$) was used as this formula best corrected for heart rate[44]. Using the intensive PK data, a simultaneous PK-QTcB model was developed, using the same procedures as described above. Linear and Emax models were tested to describe relationships between time-varying PPQ concentration and QTcB. Age, sex, and weight were tested as a covariate for the PK-QTc model.

**Associations between drug levels and drug resistance markers**. A PK-resistance genotype model was developed to quantify relationships between PPQ concentration at the time of *P. falciparum* parasitemia and wild-type or mutant genotype at *pfmdr1* N86Y, *pfmdr1* Y184F, *pfmdr1* D1246Y, and *pfcrt* K76T. In studies in Africa some or all of these mutations have been (a) associated with decreased sensitivity to chloroquine and amodiaquine and (b) selected by chemoprevention with PPQ[30,52]. The genotypes of the first new episode of parasitemia detected after a round of DP were included in the analysis. The analysis was conducted using logistic regression, and linear and Emax relationships were explored for the relationship between PPQ concentration and the probability of detecting a mutant parasite.

**Reporting summary**. Further information on research design is available in the Nature Research Reporting Summary linked to this article.

## Data availability
The raw clinical, malaria outcomes, and pharmacokinetic data used in this study have been deposited in databases available at https://doi.org/10.5281/zenodo.5602139. The data generated in this study for the figures that use model-generated data are provided in the Source Data file. Source data are provided with this paper.

## Code availability
The code used for these analyses is available at https://doi.org/10.5281/zenodo.5562807.

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

## Acknowledgements

We would like to acknowledge the study participants, clinical investigators, and study staff for the parent study, and the UCSF Drug Research Unit. The following funding from the National Institutes of Health supported this work: Institutes of Health: KL2 TR001870-04 (E.W.), AI117001 (P.J.R. and F.A.).

## Author contributions

E.W., R.M.S., P.R., F.A., G.D., and M.R.K. conceived the research. A.K., M.K.M., B.O., M.W., L.H., E.W., M.D., and J.L. collected and generated data. E.W., A.M.A., E.H., and P.J., conducted the data analysis. E.W. wrote the paper. All authors provided critical edits and approved the final version of the paper.

## Competing interests

The authors declare no competing interests.
