## [Peer Review File · Nature Communications]

Identifying an optimal dihydroartemisinin-piperaquine dosing regimen for malaria prevention in young Ugandan childrenEditorial Note: Parts of this Peer Review File have been redacted as indicated to remove third-party material where no permission to publish could be obtained.

Reviewers' Comments:

Reviewer #1:

Remarks to the Author:

The authors have presented a very well written manuscript describing a study of intermittent preventive treatment (IPT) with dihydroartemisinin-piperaquine (DP). They have identified an optimal dosing regimen associated with 95% protective efficacy in this pediatric group. The proposed age based regimen is expected to provide better efficacy than currently recommended regimens and should be simpler to employ. The data and methods used in the study are adequate for the goals and conclusions of the work.

A significant contribution of the paper is the identification of the target concentration associated with a 95% reduction in the malaria hazard. Also, the influence of malnutrition is evaluated and discussed.

Minor editorial comments for the authors:

Abstract line 26 - the phrase ", compared to every 12 weeks." isn't connected in thought to the first part of the sentence that addresses the 95% protective efficacy of the q4wk regimen. Perhaps it would be of benefit to state more clearly that q12wk is not associated with 95% protective efficacy if that is the case and the intent here.

Line 130 - "establishing" should be "established".

Line 175 - Can a statement be made regarding why Bazett's correction was chosen as the method for heart rate correction of QT? Were any of the assumptions of the Garnett White paper tested? For example was a plot of QTcB versus RR evaluated to determine if the correction adequately controls for HR effect on QTc?

Line 369 - Are higher concentrations expected to increase or decrease the risk?

Line 148 in the supplemental figure section: Figure S6 doesn't define the gold versus purple boxes. Malnourished and nourished?

Reviewer #2:

Remarks to the Author:

The authors report a PK/PD analysis based on what they describe as the largest longitudinal clinical trial of IPT with dihydroartemisinin-piperaquine (DP) in young children, including longitudinal pharmacokinetic, incident malaria, and electrocardiographic and drug sensitivity data. A piperaquine concentration of 15.4 ng/mL reduced the malaria hazard by 95%, which provides a useful target concentration for dose optimization, and is consistent with previously published IPT targets between 10 and 20 ng/mL. They have identified that children aged 1-2 years had lower piperaquine exposure than infants and are at the highest risk of malaria in the cohort. Malnourished young children were particularly at risk of low piperaquine exposure and had a 42% increased incidence of malaria, when compared with adequately nourished children. Importantly, the authors predicted that age-based (rather than weight based) 4-weekly DP dosing would provide maximal protection and reduce disparities in piperaquine exposure due to age and nutritional status. However, these age-based DP dosing strategies did not fully eliminate the impact of malnutrition on piperaquine exposure, and adherence with 3-day IPT regimens remains a major obstacle. While age-based dosing is expected to have clear operational advantages, it seems from SMC experience that the benefits will not persist if dosing intervals are extended even slightly; malaria incidence reached levels similar to no chemoprevention by 29-35 days after a round of SMC.

The findings regarding reduced piperazine exposure and increased malaria incidence in malnourished young children is of great public health significance given that one-third of children under 5 years of age in Sub-Saharan Africa are malnourished. However, they define malnourished children as those with a weight-for age z-score (WAZ) ≤ -2 at the start of chemoprevention (8 weeks of age). Nutritional status is likely to change significantly over the three years of follow up, and determinants of WAZ-scores at 8 weeks of age (when maternal health and breast feeding are key) are not the same as those in toddlers. It would be much more useful if the WAZ-scores used in this analysis were those measured at the time of DP dosing. Furthermore, the effects of malnutrition on drug exposure and malaria incidence differ significantly between children who are wasted and who are stunted. Thus weight-for-height z-scores and mid-upper arm circumference would be better measures of the complex relationships between nutritional status and antimalarial drug exposure and malaria incidence than the WAZ-scores used (or the height for age z-score (HAZ) assessed in the covariate model building). WHZ data are available for this study and should be used unless the WAZ score used is clearly justified.

It is not surprising that the authors did not find associations between piperazine concentrations and the resistance markers studied (pfmdr1 N86Y, pfmdr1 Y184F, pfmdr1 D1246Y and pfcr K76T). Firstly, drug pressure is exerted more at a community level than an individual level. The use of DP for IPT by 280 young children, of whom only 96 received 4-weekly DP-IPT is unlikely to exert substantial drug pressure on the *P. falciparum* parasites in that community. Secondly, clinically significant piperazine resistance is associated independently with plasmepsin2/3 amplification and four mutations in the pfcr gene (Thr93Ser, His97Tyr, Phe145Ile, and Ile218Phe), rather than the resistance markers pfcr 76T and pfmdr1 genes studied. The markers studied were selected based on a 2007 manuscript [Ref 21] by some of the same authors, which was on molecular markers mediating resistance to amodiaquine not piperazine. Thus, the sections on the associations between drug concentrations and molecular markers of drug resistance need to be better justified or excluded from this manuscript.

It is unclear why children born from mothers who received DP for IPT during pregnancy were randomized to either DP every 4 or 12 weeks, while children born from mothers who received SP, only received DP every 12 weeks. The authors should justify this design and discuss its implications.

MINOR COMMENTS

Introduction:

- Line 45: "approved" IPT regimen – approved by whom?
- Line 56: it would be more informative to quantify the "higher peak concentrations" associated with QTc prolongation.

Methods

- Line 83: specify whether or not the placebo was matched.
- Line 89: Discuss the matrix effects between capillary and venous samples used plasma piperazine concentrations, specify the conversion method used to pool these concentration data for the PK/PD analyses [only their log linear relationship is reported (line 211-212) and shown in Figure S2] and explain how the confounding effects of age were adjusted for in deriving this conversion method, given that different sample matrices were used for different ages in the cohort.
- Line 113-114: It is unclear whether or how data from the qualitative interpretation of chromatographs to further evaluate the samples below the lower limit of quantification [n=578 (12.6%)] conducted by two experienced mass spectrometrists was used. Please clarify.
- Line 130 should read: we first established (not establishing)
- Line 150-151: An incident malaria episode was defined as "fever and positive blood smear >14 days from a prior episode of malaria (to minimize effect of antimalarial treatment failure)". However, most PCR-confirmed treatment failures occur after 14 days, and as a full treatment dose of DP is given with each IPT administration, treatment failures before 14 days would be at least as important from a

clinical / public health perspective, although not classically defined incident malaria.

- Line 175: The use of QTcB rather than QTcF (or study specific correction of QT intervals) should be justified briefly.

Results

- Line 193: The median (range) number of piperazine concentrations per participant should be separated for those participating in intensive and sparse sampling
- Line 214: Specify cytochrome P450 iso-enzyme/s that metabolise piperazine, as their maturation with age differs.
- Line 231 (discussed in lines 351-353): Could drug interactions with concomitant medications partly explain the lower piperazine concentrations observed in some participants?
- Line 237: Why was maternal socio-economic status not included in the final model, given that it was a significant covariate for malaria hazard?

Discussion

- Line 395: The claim that there is no known artemisinin resistance in Uganda is contradicted by Asua et al (J Infect Dis 2021;223(6):985-994) who report that the prevalence of P. falciparum kelch mutations 469Y and 675V, which are associated with delayed parasite clearance, has increased at multiple sites in northern Uganda (up to 23% and 41%, respectively). As the Asua et al manuscript includes some of the authors of the manuscript under review, this would suggest that not all co-authors had reviewed the final version of the manuscript prior to submission. As artemisinin resistance is now being reported in Africa, could the authors also discuss the implications of the optimised regimen in a setting where artemisinin-resistance is being reported and potentially increasing?
- Line 415: "piperazine exposure was lower in older infants" contradicts the results reported elsewhere that this was lower in children aged 1-2 years (as infants are defined as those aged <12 months).

Tables and Figures

- Figure S6: Repeat explanation or legend to explain the difference between orange and purple bars.

Reviewer #3:

Remarks to the Author:

Identifying an optimal dihydroartemisinin-piperazine dosing regimen for malaria prevention in young Ugandan children

Erika Wallender, et al.

The authors report on an important PK/PD study of dihydroartemisinin-piperazine when used as intermittent preventive treatment. Overall, the study is conducted to a high standard and the modelling is appropriate. The main conclusions of the study are appropriate and supported by the results. The most interesting/novel aspect of the work is the age-based dosing presented in the manuscript.

I have a few specific comments that needs clarification.

Line 109: Please provide some additional details on the drug quantification method. What lab analysed drug concentrations and on what equipment? Were samples shipped there (what conditions) or analysed directly on collection? Did the authors use quality control samples to ensure accuracy and precision during quantification of clinical samples?

Line 133: Did the authors use separate residual errors for capillary and venous samples?

Line 140: why was weight evaluated as a covariate when already incorporated a priori as an allometric function?

Was MUAC available for covariate evaluation, as this has been shown to correlate with PK parameters for other drugs?

Line 144: I believe M1 and M6 are also rather common methods to handle BQL data, were these also evaluated?

Line 153: How was high intensity period incorporated as a covariate (considering its neither continuous nor categorical). In past publications, this has been modelled as a surge function on the hazard of infection.

Line 162: Please provide some more information regarding the simulations. How many simulations were performed and how were these evaluated?

Line 178: what covariates were evaluated in the PK-QTc analysis?

Line 212: why was the M2 method selected for BQL data in the final model?

Line 218/Eq. 1: where is 57 weeks coming from in equation 1? Furthermore, this maturation function will increase clearance indefinitely, with substantial increases post 2 years of life (which is not biologically plausible). It is of course important to have a biologically plausible enzyme function so that readers can extrapolate and simulate for the population in need of IPT (i.e. <5 years). An enzyme maturation function is commonly parameterised as $X=Y^z/(T_{50}^z+y^z)$ to estimate when a fully induced enzyme function is reached. Please elaborate and motivate the function used here.

Line 225/Eq. 2: This equation specifies the impact of weight-for-age (WAZ) but it is not mentioned if height-for-age (HAZ) had a similar impact, or how the impact between these two variables were evaluated. I assume they are correlated but they have fundamentally different interpretations, i.e. stunting (low height for age), wasting (low weight for height), underweight (low weight for age). Please also explain why +0.5 was added to the individual WAZ in this function.

Adherence was incorporated as a categorical effect of self-administration on F in this equation, resulting in an average 61% lower F if DP was administered at home. Did the authors consider implementing a mixture-model to better characterise this behaviour of adherence/non-adherence?

Line 242/Eq. 3: The impact of piperazine on the hazard of having a malaria episode is parameterised as an instantaneous effect, i.e. piperazine concentrations at the time of detection of malaria. However, one could argue that there is a substantial lag between the start of the blood stage infection (parasites emerging from the liver) and detection of malaria. There are a few attempts in the literature to take this into account (Bergstrand et al, *Sci Trans Med*, 2014; Chotsiri et al, *Nat Commun*, 2019). With this in mind, please comment on the approach chosen and if the estimated piperazine cut-off levels really represent the concentration needed to reduce hazard with 95%.

Line 252: The authors state "this covariate did not improve the model fit by visual predictive plot, so it was not included in the final model.". Please elaborate on this statement so that the reader understands fully why this covariate was disqualified.

Line 272: This section is a bit difficult to understand. Would be good to have a clear take-home message, e.g. simulate the trial 1000-times and visualise malaria episodes over time, associated with each of the evaluated regimens.

Furthermore, asymptomatic malaria breakthrough will happen close to the next round of treatment, and these episodes will most likely be treated successfully by the subsequent 3-day regimen. Thus, what is the difference in clinical malaria between the different regimens?

It would also strengthen the work by using a larger cohort of patients (weight and age data) to simulate different regimens and compare outcomes in a larger IPT population (<5 years of age).

Table 2: It would increase the interpretation of the data by presenting IIV as %CV, and to add RSE (from the bootstrap).

Figure S4: change y-axis to mSec

Reviewer #1:

The authors have presented a very well written manuscript describing a study of intermittent preventive treatment (IPT) with dihydroartemisinin-piperazine (DP). They have identified an optimal dosing regimen associated with 95% protective efficacy in this pediatric group. The proposed age based regimen is expected to provide better efficacy than currently recommended regimens and should be simpler to employ. The data and methods used in the study are adequate for the goals and conclusions of the work.

A significant contribution of the paper is the identification of the target concentration associated with a 95% reduction in the malaria hazard. Also, the influence of malnutrition is evaluated and discussed.

Response: Thank you for these comments.

Minor editorial comments for the authors:

1) Abstract line 26 - the phrase ", compared to every 12 weeks." isn't connected in thought to the first part of the sentence that addresses the 95% protective efficacy of the q4wk regimen. Perhaps it would be of benefit to state more clearly that q12wk is not associated with 95% protective efficacy if that is the case and the intent here.

Response: This sentence has been rearranged to clarify the comparison as follows: "Compared to DP every 12 weeks, DP every 4 weeks was associated with 95% protective efficacy (95% CI: 84-99%)." (page 2, line 25)

2) Line 130 - "establishing" should be "established".

Response: This has been corrected.

3) Line 175 - Can a statement be made regarding why Bazett's correction was chosen as the method for heart rate correction of QT? Were any of the assumptions of the Garnett White paper tested? For example was a plot of QTcB versus RR evaluated to determine if the correction adequately controls for HR effect on QTc?

Response: We used Bazett's QTc correction because there was less of a correlation between the RR interval and QTc, compared to the Fredirica correction. Also, Bazett's is the most commonly used correction for young children. As a comparison of the Frediricia and Bazett's correlations for this cohort has already been published by our group (Whalen, et al, CPT, 2019, see Figure R1 below which is from the supplement in the Whalen paper), we modified the text and cited this paper as described below.

We added the italic text to the manuscript: "The corrected QTc by Bazett's formula (QT/\sqrt{RR} , QTcB) was used *as this formula best corrected for heart rate* (45)." (Page 23, line 495)

[redacted]

Figure R1. Correlation between RR interval and QTc using the Bazett or Frederica correction.

4) Line 369 - Are higher concentrations expected to increase or decrease the risk?

Response: Thank you for this comment, we have clarified this statement as follows in italics: “We expected *sub-protective* PPQ concentrations to increase the risk of detection of more drug resistant parasites when infections occur, as has been observed for pregnant women in Uganda.” (Page 15, line 302)

5) Line 148 in the supplemental figure section: Figure S6 doesn't define the gold versus purple boxes. Malnourished and nourished?

Response: Thank you for noting this, we added the following to Figure S6 legend: “The gold boxes indicate a malnourished population at the start of chemoprevention with $WAZ \leq -2$ and purple indicates a better nourished population with $WAZ > -2$ at the start of chemoprevention.”

Reviewer #2 (Remarks to the Author):

The authors report a PK/PD analysis based on what they describe as the largest longitudinal clinical trial of IPT with dihydroartemisinin-piperaquine (DP) in young children, including longitudinal pharmacokinetic, incident malaria, and electrocardiographic and drug sensitivity data. A piperaquine concentration of 15.4 ng/mL reduced the malaria hazard by 95%, which provides a useful target concentration for dose optimization, and is consistent with previously published IPT targets between 10 and 20 ng/mL. They have identified that children aged 1-2

years had lower piperazine exposure than infants and are at the highest risk of malaria in the cohort. Malnourished young children were particularly at risk of low piperazine exposure and had a 42% increased incidence of malaria, when compared with adequately nourished children. Importantly, the authors predicted that age-based (rather than weight based) 4-weekly DP dosing would provide maximal protection and reduce disparities in piperazine exposure due to age and nutritional status. However, these age-based DP dosing strategies did not fully eliminate the impact of malnutrition on piperazine exposure, and adherence with 3-day IPT regimens remains a major obstacle. While age-based dosing is expected to have clear operational advantages, it seems from SMC experience that the benefits will not persist if dosing intervals are extended even slightly; malaria incidence reached levels similar to no chemoprevention by 29-35 days after a round of SMC.

6) The findings regarding reduced piperazine exposure and increased malaria incidence in malnourished young children is of great public health significance given that one-third of children under 5 years of age in Sub-Saharan Africa are malnourished. However, they define malnourished children as those with a weight-for age z-score (WAZ) ≤ -2 at the start of chemoprevention (8 weeks of age). Nutritional status is likely to change significantly over the three years of follow up, and determinants of WAZ-scores at 8 weeks of age (when maternal health and breast feeding are key) are not the same as those in toddlers. It would be much more useful if the WAZ-scores used in this analysis were those measured at the time of DP dosing.

Response: Thank you for this comment. We regret that this information was not clear. Fortunately, we implemented the WAZ covariate exactly as the reviewer recommended and time-varying z-scores were used for all steps of model development and simulations. We used the 8-weeks of age WAZ stratification to display simulation results only. Based on the reviewer's input, we have modified simulation results to stratify results by the WAZ at the time of each dosing interval. We made the following modifications to the text and figures:

- We added the term "time-varying" before all instances when z-score covariates were described in the methods section. (page 21, line 449 & page 22, line 464)

- We revised our summary of the simulation results in **Table 3** and **Figure 6B** to group the nutritional status of children by the WAZ at the time of the most recent DP dosing event. In addition, for **Table 3**, we added columns to display the percentage of dosing events in the simulations where a child maintained a protective concentration >15.4 ng/mL piperazine for >80% of the time to better represent the distribution of data from the simulation results.

Table 3. Time above protective PPQ concentrations by adherence status and weight for age z-score at the time of most recent DP course.

Regimen	% time above protective PPQ levels, (2.5-97.5% of population)								
	Full adherence			2/3 adherence			1/3 adherence		
	WAZ ≤-2	WAZ >-2	Percentage of children with >80% time above 15.4 ng/mL	WAZ ≤-2	WAZ >-2	Percentage of children with >80% time above 15.4 ng/mL	WAZ ≤-2	WAZ >-2	Percentage of children with >80% time above 15.4 ng/mL
Every 4-week DP									
Clinical trial protocol	69.5 (14.6-100)	93.1 (19.4-100)	57.3 (54.8-59.5)	36.7 (8.7-100)	56.5 (11.8-100)	32.8 (30.9-34.8)	16.0 (3.9-95.9)	21.9 (5.3-100)	9.7 (8.6-11.0)
WHO 2015	88.1 (18.8-100)	100 (30.2-100)	75.6 (73.4-77.4)	53.1 (11.5-100)	87.5 (17.6-100)	52.2 (50.1-54.5)	21.2 (5.1-100)	37.3 (7.9-100)	21.1 (19.5-22.8)
Proposed age-based	100 (30.3-100)	100 (36.5-100)	82.2 (80.2-83.9)	83.1 (18.0-100)	95.2 (20.6-100)	60.1 (57.9-62.2)	36.1 (8.9-100)	45.0 (9.3-100)	26.3 (24.4-28.2)
Every 8-week DP									
Clinical trial protocol	20.8 (6.2-81.2)	28.9 (7.8-97.0)	6.7 (5.6-7.9)	11.9 (3.9-57.0)	15.8 (5.0-74.1)	1.7 (1.2-2.2)	6.3 (1.8-26.3)	7.9 (2.4-39.1)	.1 (<.01-.3)
WHO 2015	27.6 (7.6-93.2)	46.6 (11.5-100)	16.2 (14.5-17.9)	15.3 (4.9-	26.7 (7.3-94.7)	5.5 (4.5-6.5)	7.7 (2.3-37.2)	11.9 (3.5-62.1)	.7 (.4-1.0)

				70.6)					
Proposed age-based	44.4 (11.6-100)	51.9 (12.9-100)	19.9 (17.9-21.8)	25.7 (7.3-89.1)	30.0 (8.1-96.6)	7.0 (5.9-8.2)	11.7 (3.5-57.2)	13.2 (3.9-65.6)	.9 (.5-1.3)

WAZ = weight-for-age z-score

Clinical trial protocol: <6 kg: DHA/PPQ 10/80 mg daily x 3 days, 6-<11 kg: DHA/PPQ 20/160 mg daily x 3 days, 11-<15 kg:

DHA/PPQ 30/240 mg daily x 3 days, 15-<20 kg: DHA/PPQ 40/320 mg daily x 3 days

WHO 2015: <8 kg: DHA/PPQ 20/160 mg daily x 3 days, 8-<11 kg: DHA/PPQ 30/240 mg daily x 3 days, 11-<17 kg: DHA/PPQ 40/320 mg daily x 3 days, 17-<25 kg: DHA/PPQ 50/480 mg daily x 3 days

Proposed age-based: 2-6 months of age: DHA/PPQ 20/160 mg daily x 3 days, 6-18 months of age: DHA/PPQ 30/240 mg daily x 3 days, 18-24 months of age: DHA/PPQ 40/320 mg daily x 3 days

Figure 6. Simulation results. (A) Predicted PPQ trough concentrations for simulated DP regimens, stratified by nutritional status. The red line indicates the 15.4 ng/mL PPQ target. Boxes indicate 25-75% of population incidence, and vertical bars represent 95% of the population. (B) Predicted malaria incidence by DP regimen with increasing baseline malaria transmission, stratified by nutritional status and adherence level (1/3 adherence indicates bioavailability observed for non-directly observed therapy in the study, 2/3 adherence indicates a bioavailability midpoint between the directly and non-directly observed population, and full adherence indicates the bioavailability observed in the directly observed therapy group).

7) Furthermore, the effects of malnutrition on drug exposure and malaria incidence differ significantly between children who are wasted and who are stunted. Thus weight-for-height z-scores and mid-upper arm circumference would be better measures of the complex relationships between nutritional status and antimalarial drug exposure and malaria incidence than the WAZ-scores used (or the height for age z-score (HAZ) assessed in the covariate model building). WHZ data are available for this study and should be used unless the WAZ score used is clearly justified.

Response: Thank you for this comment. Unfortunately, mid upper arm circumference data was not collected in this trial, so it could not be evaluated. WAZ, HAZ, and WHZ were all tested as potential covariates in the model for the clearance, volume and apparent bioavailability parameters. WAZ was correlated with HAZ and WHZ (see Figure R2 below). WAZ provided the greatest statistical significance and covariate effect size among markers of malnutrition for bioavailability (Δ OFV WAZ -22.1, 10.2% lower bioavailability per 1 SD decrease in z-score), compared to HAZ and WHZ (Δ OFV HAZ -5.93, 4.7% decrease in bioavailability per 1 SD decrease in z-score; Δ OFV WHZ -6.47, 4.4% decrease in bioavailability per 1 SD decrease in z-score). In addition, the final model with WAZ as a covariate for bioavailability provided an excellent visual predictive check in our data. As a result, we used WAZ in the final model. WAZ has been associated with decreased oral bioavailability for lumefantrine (Chostri, et al, Clin Pharmacol Ther, 2019) and for sulfadoxine and pyrimethamine (de Kock, M, et al, AAC, 2018) and with lower day 7 lumefantrine levels for the same mg/kg dose for lumefantrine. With a literature precedent and our model fit, WAZ was a valuable covariate in our model. We agree with the reviewer that further assessment of this complex relationship between different markers of malnutrition and PK exposure is warranted.

- We significantly revised the PK section of the results to better explain the covariate selection for malnutrition as follows: “A higher degree of malnutrition, as measured by either lower weight for age z-score (WAZ), height for age z-score (HAZ), or weight for height z-score (WHZ), all reduced PPQ bioavailability (Δ OFV WAZ -22.1, Δ OFV HAZ -5.93, Δ OFV WHZ -6.47). Malnutrition, as measured by WAZ provided the greatest statistical significance, the greatest covariate effect size (for each SD decrease in z-score there was a decreased bioavailability by 10.2% for WAZ vs 4.7% for HAZ or 4.4% for WHZ), and the best model fit by visual predictive check. Therefore, WAZ was selected as the covariate to represent malnutrition in the final PK model.” (page 6, line 112)
- We also added the following comment in italics to the discussion: “Lower oral bioavailability has been linked to malnutrition, *as defined by low WAZ*, among children <5 years of age for lumefantrine *and SP (18, 19). A variety of biomarkers of acute malnutrition including mid-upper arm circumference (not available for this study), WHZ, and WAZ have been linked to lower antimalarial drug exposure for malaria treatment. More research is needed to elucidate the pathophysiologic mechanism for these findings (18, 20, 21).*” (page 12, line 249)

Figure R2. Correlation between weight for age z-score and (A) height for age z-score and (B) weight for length z-score.

8) It is not surprising that the authors did not find associations between piperazine concentrations and the resistance markers studied (pfmdr1 N86Y, pfmdr1 Y184F, pfmdr1 D1246Y and pfcr1 K76T). Firstly, drug pressure is exerted more at a community level than an individual level. The use of DP for IPT by 280 young children, of whom only 96 received 4-weekly DP-IPT is unlikely to exert substantial drug pressure on the *P. falciparum* parasites in that community.

Secondly, clinically significant piperazine resistance is associated independently with plasmepsin2/3 amplification and four mutations in the pfcr1 gene (Thr93Ser, His97Tyr, Phe145Ile, and Ile218Phe), rather than the resistance markers pfcr1 76T and pfmdr1 genes studied. The markers studied were selected based on a 2007 manuscript [Ref 21] by some of the same authors, which was on molecular markers mediating resistance to amodiaquine not piperazine. Thus, the sections on the associations between drug concentrations and molecular markers of drug resistance need to be better justified or excluded from this manuscript.

Response: The reviewer highlights the complexity of relationships between antimalarial drug exposure and drug resistance selection. We argue that the genetic markers included in this report (pfmdr1 86, 1246 and pfcr1 76) remain relevant in Uganda, because they have been associated with decreased aminoquinoline sensitivity in Uganda by our group and others in sub-Saharan Africa (see citations in the next section of this response). In addition, we have demonstrated in our own work on malaria chemoprevention with DP that higher piperazine concentrations at the time of parasitemia were associated with an increased risk of detecting a pfmdr1 86Y or pfcr1 76T mutations at the time of parasitemia among pregnant women (Conrad, et al JID, 2017 and Wallender, et al, AAC, 2019).

The reviewer provided a helpful summary of PPQ resistance markers that are important in SE Asia. Fortunately, to date there is no convincing evidence of PPQ resistance mediated by the

noted pfcrt mutations or plasmepsin2/3 copy number amplification in African *P. falciparum*. Work from our group and others has shown more modest decreases in PPQ potency mediated by the mutations studied in our report. Our study was conducted based on our strong understanding of drug resistance in Uganda and our prior evaluations of drug resistance markers in the setting of IPT with DP in Uganda (Asua, V, et al, JID, 2020, Rasmussen, SA, et al, AAC, 2017, Conrad, M, et al, JID, 2017, Wallender, et al, AAC, 2019).

We added the following to relevant text to the methods section: “In studies in Africa some or all of these mutations have been a) associated with decreased sensitivity to chloroquine and amodiaquine and b) selected by chemoprevention with PPQ (31, 53).” (page 23, line 504)

9) It is unclear why children born from mothers who received DP for IPT during pregnancy were randomized to either DP every 4 or 12 weeks, while children born from mothers who received SP, only received DP every 12 weeks. The authors should justify this design and discuss its implications.

Response: We added the following text in italics to clarify the randomization structure, which was not determined by the lead investigators for the PK/PD study.

- 1) “Children born from mothers who received DP for IPT during pregnancy were randomized to either DP every 4 or 12 weeks, whereas children born from mothers who received SP *were all randomized to IPT with DP every 12 weeks in order to maximize the power of the parent study to detect differences in malaria incidence in childhood resulting from the IPT regimen received during pregnancy.*” (page 18, line 385)
- 2) “*Maternal chemoprevention regimen was not statistically significantly associated with PK exposure.*” (page 7, line 139)

MINOR COMMENTS

Introduction:

10) Line 45: “approved” IPT regimen – approved by whom?

Response: We added “a highly effective WHO or country approved” to this line. (page 3, line 45)

11) Line 56: it would be more informative to quantify the “higher peak concentrations” associated with QTc prolongation.

Response: Thank you for this comment, we revised the statement as follows: “Large studies have not identified significant clinical toxicities associated with IPT with DP in children, even though *plasma PPQ concentration has been positively associated with lengthening of the corrected QT interval (QTc) (10, 11).*” (page 4, line 57)

Methods

12) Line 83: specify whether or not the placebo was matched.

Response: We added “matched” to this statement.

13) Line 89: Discuss the matrix effects between capillary and venous samples used plasma piperazine concentrations, specify the conversion method used to pool these concentration data for the PK/PD analyses [only their log linear relationship is reported (line 211-212) and shown in Figure S2] and explain how the confounding effects of age were adjusted for in deriving this conversion method, given that different sample matrices were used for different ages in the cohort.

Response: We obtained matched capillary and venous plasma from children across the full age range in the study. In total, we had 133 paired venous/capillary samples from 70 children of ages 15 to 104 months. In addition to graphical examination, we evaluated the fit of different functions within the PK/PD model, where we found the log-linear relationship provided the best fit, as suggested by the raw data. We added the following phrase in italics: “A log-linear relationship best described the relationship between venous and capillary concentrations, *and the relationship did not vary by age during the study period (Figure S3).*” (page 5, line 98)

14) Line 113-114: It is unclear whether or how data from the qualitative interpretation of chromatographs to further evaluate the samples below the lower limit of quantification [n=578 (12.6%)] conducted by two experienced mass spectrometrists was used. Please clarify.

Response: We clarified this section as follows in italics: “Qualitative interpretation of chromatographs to further evaluate BLQ samples was conducted by two experienced mass spectrometrists. BLQ samples with no observed peak, *based on review by the mass spectrometrists*, or which were measured after a prior BLQ measurement without interval dosing of DP were excluded from the analysis. *If a peak was observed in an BLQ samples, it was considered as BLQ.*” (page 20 line 420)

15) Line 130 should read: we first established (not establishing)

Response: This has been corrected.

16) Line 150-151: An incident malaria episode was defined as “fever and positive blood smear >14 days from a prior episode of malaria (to minimize effect of antimalarial treatment failure)”. However, most PCR-confirmed treatment failures occur after 14 days, and as a full treatment dose of DP is given with each IPT administration, treatment failures before 14 days would be at least as important from a clinical / public health perspective, although not classically defined incident malaria.

Response: Thank you for this comment. By removing malaria episodes <14 days after a prior malaria diagnosis, we minimized the antimalarial effect of concurrent use of artemether-lumefantrine on the PK/PD relationship. If malaria was diagnosed within 14 days of receiving IPT with DP, however, it was included in the analysis. We added the following to the text in italics to clarify:

- “*Uncomplicated malaria was treated with artemether-lumefantrine.*” (Page 21, Line 461)
- “An incident malaria episode was defined as fever and positive blood smear >14 days from a prior episode of malaria (to minimize effect of *artemether-lumefantrine* treatment).” (Page 21, Line 448)

17) Line 175: The use of QTcB rather than QTcF (or study specific correction of QT intervals) should be justified briefly.

Response (as described in response to reviewer 1, response #3): We used Bazett’s QTc correction because there was less of a correlation between the RR interval and QTc, compared to the Fredirica correction. Also, Bazett’s is the most commonly used correction for young children. As a comparison of the Frediricia and Bazett’s correlations for this cohort has already been published by our group (Whalen, et al, CPT, 2019, see Figure R1 below which is from the supplement in the Whalen paper), we modified the text and cited this paper as described below.

We added the italic text to the manuscript: “The corrected QTc by Bazett’s formula (QT/\sqrt{RR} , QTcB) was used *as this formula best corrected for heart rate* (45).” (Page 23, line 495)

[redacted]

Figure R1. Correlation between RR interval and QTc using the Bazett or Frederica correction.

Results

18) Line 193: The median (range) number of piperazine concentrations per participant should be separated for those participating in intensive and sparse sampling

Response: We made the following change to the text: All participants had at least one PPQ concentration determined (median number [range] per participant: 31 [16-33] for intensive PK sampling; 12 [1-20] for sparse PK sampling). (page 5, line 77)

19) Line 214: Specify cytochrome P450 iso-enzyme/s that metabolise piperazine, as their maturation with age differs.

Response: We revised cytochrome P450 to cytochrome P450 3A4 (Page 6, Line 106)

20) Line 231 (discussed in lines 351-353): Could drug interactions with concomitant medications partly explain the lower piperazine concentrations observed in some participants?

Response: Thank you for this comment. Potentially interacting medications were avoided during the study. We added the following to the methods section: “Concurrent use of other medications with antimalarial activity or which could interact with DP (cytochrome 3A4 inducers or inhibitors) were avoided.” (Page 19, line 397)

21) Line 237: Why was maternal socio-economic status not included in the final model, given that it was a significant covariate for malaria hazard?

Response: Thank you for this comment. We were not able to include SES in the final model because of overall model instability. When we attempted to incorporate SES as a covariate in the final model, we observed problems with obtaining convergence with the full data set and we were unable to reliably determine the confidence intervals around preliminary parameter estimates using the bootstrap method. We added the following to the results and discussion, respectively:

- 1) Results section: “Maternal SES, as defined by a propensity score summarizing property and income, was assigned a value between -1 and 3. In univariate analysis, we found that each 1 unit increase in maternal SES was associated with a 26.2% decreased risk of malaria (Δ OFV -7.21). However, when we incorporated SES into the full PK/PD model we encountered unacceptable model instability and confidence intervals could not be reliably acquired by bootstrap, so maternal SES was not included in the final model.” (page 8, line 158)
- 2) Discussion: “Finally, we could not include maternal SES as a covariate on the baseline malaria hazard in the final PK/PD due to model instability. Fortunately, though SES status reduced the intraindividual variability in our exploratory models, it did not modify the key PK/PD relationships or baseline malaria hazard estimates during

chemoprevention. Future studies should consider an externally validated SES measure.”
(page 17, line 352)

Discussion

22) Line 395: The claim that there is no known artemisinin resistance in Uganda is contradicted by Asua et al (J Infect Dis 2021;223(6):985-994) who report that the prevalence of *P. falciparum* kelch mutations 469Y and 675V, which are associated with delayed parasite clearance, has increased at multiple sites in northern Uganda (up to 23% and 41%, respectively). As the Asua et al manuscript includes some of the authors of the manuscript under review, this would suggest that not all co-authors had reviewed the final version of the manuscript prior to submission.

Response: Unfortunately, the paper cited by this reviewer from our group did not evaluate clinical resistance to artemisinins. Setting up a clinical trial in northern Uganda during the pandemic has been problematic, but we are working to characterize the parasitological and clinical implications of our findings as soon as possible. However, in Asua, V, et al, we showed an increased prevalence of two K13 mutations that might mediate delayed parasite clearance after treatment with artemisinins at some sites in northern Uganda. Fortunately, we did conduct surveillance in SE Uganda, where the study discussed in this manuscript took place, and we did not find the relevant K13 mutations. For a better understanding of regional drug resistance situation, we suggest our definitive review on antimalarial drug resistance in Africa from our group (Conrad, et al, Lancet ID, 2019).

To clarify this point for readers, we add the following text in italics: “To incorporate the parasitocidal activity of dihydroartemisinin into our model, we assumed cumulative survival returned to 100% when DP was given, *as dihydroartemisinin resistance has yet to be detected in southeastern Uganda at the time of the study (36, 37).*” (page 16, line 333)

23) As artemisinin resistance is now being reported in Africa, could the authors also discuss the implications of the optimised regimen in a setting where artemisinin-resistance is being reported and potentially increasing?

Response: Thank you for this comment and opportunity. There is general consensus (e.g. discussed at recent meetings sponsored by WHO, WWARN, and MMV) that the most convincing evidence for artemisinin delayed clearance (often referred to as resistance) is from Rwanda. In Rwanda, they found delayed clearance after therapy in patients who nonetheless responded well to therapy with artemether-lumefantrine. There is no convincing evidence of clinical ACT resistance in Africa; a few trials suggesting ACT treatment efficacy <90% have been challenged for technical reasons (see an editorial on this topic by Rosenthal in the AJTMH, now in press).

In our discussion we address this concern by adding the following text: “Importantly, artemisinin resistance may be emerging in east Africa, supported by identification of parasites with mutations in the K13 gene known or suspected of mediating resistance in southeast Asia in Rwanda (34, 35) and Uganda (36, 37). Resistance to PPQ in southeast Asia is associated with mutations in *pfprt* and amplification of *plasmepsin* genes that have generally not been described

in Africa (38). Our study was conducted in an area without emergence of these mutations (36, 37), but in the event of resistance, we would expect that higher PPQ concentrations would be needed to prevent malaria, and that infection during sub-protective PPQ levels would select for resistance. This concern highlights the importance of minimizing breakthrough malaria infections during IPT, continued evaluation of IPT preventive efficacy, and continued surveillance for drug resistance, as is ongoing.” (Page 15, line 310)

24) Line 415: “piperazine exposure was lower in older infants” contradicts the results reported elsewhere that this was lower in children aged 1-2 years (as infants are defined as those aged <12 months).

Response: We changed older infants to “... children 1-2 years of age” (Page 17, Line 367)

Tables and Figures

25) Figure S6: Repeat explanation or legend to explain the difference between orange and purple bars.

Response: Thank you, this has been corrected.

Reviewer #3 (Remarks to the Author):

Identifying an optimal dihydroartemisinin-piperazine dosing regimen for malaria prevention in young Ugandan children

Erika Wallender, et al.

The authors report on an important PK/PD study of dihydroartemisinin-piperazine when used as intermittent preventive treatment. Overall, the study is conducted to a high standard and the modelling is appropriate. The main conclusions of the study are appropriate and supported by the results. The most interesting/novel aspect of the work is the age-based dosing presented in the manuscript.

I have a few specific comments that needs clarification.

26) Line 109: Please provide some additional details on the drug quantification method. What lab analysed drug concentrations and on what equipment? Were samples shipped there (what conditions) or analysed directly on collection? Did the authors use quality control samples to ensure accuracy and precision during quantification of clinical samples?

Response: Thank you for this comment. We have added a full description of PK sample handling in the supplementary text which reads as follows: “Plasma for PPQ concentration quantification was separated from whole blood at the time of collection, and frozen immediately for storage at -80°C. Plasma was shipped on dry ice to the UCSF Drug Research Unit in San

Francisco, CA. PPQ quantification was conducted by protein precipitation using 25 μ L of plasma, followed by liquid chromatography and tandem mass spectrometry. Calibrated standards ranging from 0.5 to 50 ng/mL were used and QC samples comprised of 1.5, 20, 40 ng/mL for sparse samples and standards from 10 to 1,000 ng/mL and QC samples comprised of 30, 200, and 800 were used for the intensive PK analysis (46).”

27) Line 133: Did the authors use separate residual errors for capillary and venous samples?

Response: Although separate errors for capillary and venous samples were tested, the data did not support separate residual errors (no improvement in the fit and similar magnitude when errors were estimated separately), therefore these were not included in the model. The PPQ plasma quantification method was the same for both types of samples.

28) Line 140: why was weight evaluated as a covariate when already incorporated a priori as an allometric function?

Response: Thank you for noting this. We removed weight from the covariate list.

29) Was MUAC available for covariate evaluation, as this has been shown to correlate with PK parameters for other drugs?

Response: Unfortunately, MUAC was not collected in the study, and so it could not be included as a covariate. We added some additional text relevant to MUAC into the discussion as follows: “Lower oral bioavailability has been linked to malnutrition, as defined by low WAZ, among children <5 years of age for lumefantrine and SP (29, 30). A variety of biomarkers of acute malnutrition including mid-upper arm circumference (not available for this study), WHZ, and WAZ have been strongly linked to lower antimalarial drug exposure for malaria treatment, and more research is needed to elucidate the pathophysiologic mechanism for these findings (29, 31, 32).” (line 12, page 248)

30) Line 144: I believe M1 and M6 are also rather common methods to handle BQL data, were these also evaluated?

Response: Thank you for noting our typo. Indeed, the M6 method was used in the final model. This has been corrected as described in comment “Line 212” (Response #34) below.

31) Line 153: How was high intensity period incorporated as a covariate (considering its neither continuous nor categorical). In past publications, this has been modelled as a surge function on the hazard of infection.

Response: Thank you for the comment. We included the high transmission periods as a time varying categorical covariate (0 for low intensity seasons, 1-3 for the three various calendar high transmission seasons which had differing degrees of transmission). We would have preferred to use a parametric distribution, such as a surge or cosine function, however this was not possible as

transmission periods were determined by calendar time rather than age and that transmission peaks differed by year for ecological reasons.

We added “categorical” to the description of the transmission period covariate (page 8, line 144)

32) Line 162: Please provide some more information regarding the simulations. How many simulations were performed and how were these evaluated?

Response: Thank you for this comment. We added the following passage to the methods section to clarify our approach: “For each regimen and adherence pattern (1/3, 2/3 or full adherence), the final PK model was simulated 1,000 times and the final parametric survival model for malaria hazard was simulated 10,000 times using the full monthly demographic data from a combined dataset of the 280 study participants from 8 weeks to 24 months of age which data contributed to this study and from 576 children from 6 month to 24 month of age enrolled in a previously conducted set of clinical trials in Tororo, Uganda (3, 6). The percentage of time above the model-derived protective PPQ concentration per DP treatment course was estimated for each regimen from the final PK model. Simulations of the final parametric survival model were conducted using baseline hazards from 0.5 to 8 episodes per person year. Malaria hazard was kept constant in the simulations.” (page 22, line 482)

33) Line 178: what covariates were evaluated in the PK-QTc analysis?

Response: We added the following: “Age, sex and weight were tested as covariates for the PK-QTc model.” (Page 23, line 498)

34) Line 212: why was the M2 method selected for BQL data in the final model?

Response: Thank you for nothing this error on our side. We used the M6 method, but we also evaluated the M3 method, and both gave similar results. Given that 12.6% of our samples were BLQ, it is not surprising that both methods resulted in similar parameter estimates. This was clarified in the text as follows: “The BLQ/2 method (M6 method) and estimation of the likelihood of BLQ (M3 method) provided similar parameter estimates, and so the BLQ/2 method was used to handle BLQ data, with BLQ PPQ measurements assigned to 0.25 ng/mL.” (page 6, line 100)

35) Line 218/Eq. 1: where is 57 weeks coming from in equation 1?

Response: The median age from the data was 57 weeks, and this was used for the maturation function.

We added the following clarification in italics: “... and increasing age in weeks *centered on the population median of 57 weeks*, representing liver enzyme maturation during infancy ...” (page 6, line 106)

36) Furthermore, this maturation function will increase clearance indefinitely, with substantial increases post 2 years of life (which is not biologically plausible). It is of course important to have a biologically plausible enzyme function so that readers can extrapolate and simulate for the

population in need of IPT (i.e. <5 years). An enzyme maturation function is commonly parameterised as $X=Y^z/([T50] ^z+y^z)$ to estimate when a fully induced enzyme function is reached. Please elaborate and motivate the function used here.

The power function used for enzyme maturation was data driven. An Emax model provided an estimated EC50 of 15 weeks (near first time point with PK data) and the OFV was +33 compared to the model with the power function for enzyme maturation. The power and Emax functions had a similar shape during the age period, suggesting the maximum maturation was not reached. We used the function with the best overall OFV and which had been previously used for piperazine in this age group, where a similar median age of 12 months was used (Sambol, NC, et al, Clin Pharmacol Ther, 2015).

However, as the reviewer points out, this function is not appropriate to use beyond age range of the data used to build the model. Therefore, we took a conservative approach by not extrapolating beyond 2 years of age in part because of the age function, but also because WAZ was an important covariate in our model for very young children, but has not been a consistent covariate in population PK models of antimalarials for older children.

We have included the following text: “As data from children >2 years of age were not available for PK model development and age was found to impact PPQ clearance up through 2 years of age, we could not use our model to predict how PPQ parameters would be altered for older aged children and we did not conduct simulations in children older than 2 years of age. Further study of optimal DP dosing is needed for malaria chemoprevention in older age groups.” (Page 14, Line 296)

37) Line 225/Eq. 2: This equation specifies the impact of weight-for-age (WAZ) but it is not mentioned if height-for-age (HAZ) had a similar impact, or how the impact between these two variables were evaluated. I assume they are correlated but they have fundamentally different interpretations, i.e. stunting (low height for age), wasting (low weight for height), underweight (low weight for age).

Response: Thank you for bring up this issue, which was also raised by reviewer 2 in response to comment #7. WAZ, HAZ and low weight for height were correlated (Figure R2 below). We significantly revised the PK section for the results to better explain the covariate selection for malnutrition as follows: “A higher degree of malnutrition, as measured by either lower weight for age z-score (WAZ), height for age z-score (HAZ), or weight for height z-score (WHZ), all reduced PPQ bioavailability (Δ OFV WAZ -22.1, Δ OFV HAZ -5.93, Δ OFV WHZ -6.47). Malnutrition, as measured by WAZ provided the greatest statistical significance, the greatest covariate effect size (for each SD decrease in z-score there was a decreased bioavailability by 10.2% for WAZ vs 4.7% for HAZ or 4.4% for WHZ), and the best model fit by visual predictive check. Therefore, WAZ was selected as the covariate to represent malnutrition in the final PK model.” (page 6, line 112)

Figure R2. Correlation between weight for age z-score and (A) height for age z-score and (B) weight for length z-score.

38) Please also explain why +0.5 was added to the individual WAZ in this function.

Response: We added the following clarification: "... θ_{WAZ} is time-varying WAZ centered on the median value ..." and revised the equation 2 better display how the median WAZ was used. (page 12, line 246)

39) Adherence was incorporated as a categorical effect of self-administration on F in this equation, resulting in an average 61% lower F if DP was administered at home. Did the authors consider implementing a mixture-model to better characterise this behaviour of adherence/non-adherence?

Response: Thank you for this comment. We tested a mixture model in an attempt to identify participant populations within the cohort with either consistently high or low F for PPQ. We tested for the presence of 2 or 3 populations with similar bioavailability patterns over the 22-month intervention period. In these mixture models, all participants were assigned into a single population, and distinct subpopulations could not be identified. As the mixture model was unable to identify distinct subpopulations, we did not use one for the final model.

We added the following statement: "A mixture model was tested to evaluate for participant populations with either high or low apparent bioavailability, but distinct populations could not be identified." (Page 7, line 137)

40) Line 242/Eq. 3: The impact of piperazine on the hazard of having a malaria episode is parameterised as an instantaneous effect, i.e. piperazine concentrations at the time of detection of malaria. However, one could argue that there is a substantial lag between the start of the blood stage infection (parasites emerging from the liver) and detection of malaria. There are a few attempts in the literature to take this into account (Bergstrand et al, Sci Trans Med, 2014;

Chotsiri et al, Nat Commun, 2019). With this in mind, please comment on the approach chosen and if the estimated piperazine cut-off levels really represent the concentration needed to reduce hazard with 95%.

Response: Thank you for this comment. We evaluated as semi-mechanistic model which included parasite density data in the model as described Bergstrand and Chotsiri. However, when we implemented the parasite replication model estimates, which are based on malaria naïve adults, we found that the predicted windows of parasite emergence would at times overlap with DP dosing events. We were concerned that the weight-based adjustment used to scale the total body parasite burden was not sufficient for very young children. Thus, we selected the empirical model to predict clinical malaria. For our model, drug levels at the time of malaria would be correlated with drug levels when parasites emerged as all individuals received the same three day treatment course of DP and we opted to only simulate three-day DP regimens for IPT. We the following two sections to the text:

- “A semi-mechanistic model was explored which incorporated parasite replication rates extrapolated from experimental infection studies in malaria naïve adult populations (12, 13), which would enable us to predict PPQ concentrations at the time of liver emergence. We found that in our study population, the semi-mechanistic model did not predict the data well, and the empirical model was used as the final model.” (Page 8, Line 163)
- “As all participants received the same three-day treatment courses, PPQ concentrations which prevent clinical malaria would correlate with PPQ levels at liver emergence, and as a result we only simulated three-day treatment courses for our optimized regimens.” (Page 17, Line 349)

41) Line 252: The authors state “this covariate did not improve the model fit by visual predictive plot, so it was not included in the final model.”. Please elaborate on this statement so that the reader understands fully why this covariate was disqualified.

Response: Thank you for this comment. We received a similar comment from Reviewer 2, comment #21, and we have included our response and the modifications to the manuscript below.

We were not able to include SES in the final model because of overall model instability. When we attempted to incorporate SES as a covariate in the final model, we observed problems with obtaining convergence with the full data set and we were unable to reliably determine the confidence intervals around preliminary parameter estimates using the bootstrap method. We added the following to the results and discussion, respectively:

- 1) Results section: “Maternal SES, as defined by a propensity score summarizing property and income, was assigned a value between -1 and 3. In univariate analysis, we found that each 1 unit increase in maternal SES was associated with a 26.2% decreased risk of malaria (Δ OFV -7.21). However, when we incorporated SES into the full model PKPD model we encountered unacceptable model instability and confidence intervals could not be acquired by bootstrap, so maternal SES was not included in the final model. (page 8, line 158)

2) Discussion: “Finally, we could not include maternal SES as a covariate on the baseline malaria hazard in the final PK/PD due to model instability. Fortunately, though SES status reduced the intraindividual variability in our exploratory models, it did not modify the key PK/PD relationships or baseline malaria hazard estimates during chemoprevention. Future studies should consider an externally validated SES measure.” (page 17, line 352)

42) Line 272: This section is a bit difficult to understand. Would be good to have a clear take-home message, e.g. simulate the trial 1000-times and visualise malaria episodes over time, associated with each of the evaluated regimens.

Response: We added the following to this section to clarify our approach: “For each regimen, 1,000 simulations of the PK model and 10,000 simulations of the parametric survival model were conducted using longitudinal demographic data from 856 Ugandan children (280 children who contributed data to this analysis and 576 children from 6 months to 2 years of age from two prior study cohorts from the same region) (3, 6). Time above protective PPQ concentrations and clinical malaria incidence were calculated.” (Page 10, Line 189)

43) Furthermore, asymptomatic malaria breakthrough will happen close to the next round of treatment, and these episodes will most likely be treated successfully by the subsequent 3-day regimen Thus, what is the difference in clinical malaria between the different regimens?

Response: We agree that clinical malaria was most important outcome, and this was used as the primary outcome for all simulations. We added clinical malaria to the outcome description for this section to clarify this outcome.

44) It would also strengthen the work by using a larger cohort of patients (weight and age data) to simulate different regimens and compare outcomes in a larger IPT population (<5 years of age).

Response: Thank you for this comment. We have access to a longitudinal cohort of 576 young children from 6 months to 2 years of age with full monthly demographic data. We added this population to the simulation dataset used for Table 3 and Figure 6A and 6B as shown below. As previously mentioned in the response to comment #36, we avoided simulating results from children >2 years of age due to limitations in our dataset.

Table 3. Time above protective PPQ concentrations by adherence status and weight for age z-score at the time of most recent DP course. .

Regimen	% time above protective PPQ levels, (2.5-97.5% of population)								
	Full adherence			2/3 adherence			1/3 adherence		
	WAZ ≤-2	WAZ >-2	Percentage of children with >80% time above 15.4 ng/mL	WAZ ≤-2	WAZ >-2	Percentage of children with >80% time above 15.4 ng/mL	WAZ ≤-2	WAZ >-2	Percentage of children with >80% time above 15.4 ng/mL
Every 4-week DP									
Clinical trial protocol	69.5 (14.6-100)	93.1 (19.4-100)	57.3 (54.8-59.5)	36.7 (8.7-100)	56.5 (11.8-100)	32.8 (30.9-34.8)	16.0 (3.9-95.9)	21.9 (5.3-100)	9.7 (8.6-11.0)
WHO 2015	88.1 (18.8-100)	100 (30.2-100)	75.6 (73.4-77.4)	53.1 (11.5-100)	87.5 (17.6-100)	52.2 (50.1-54.5)	21.2 (5.1-100)	37.3 (7.9-100)	21.1 (19.5-22.8)
Proposed age-based	100 (30.3-100)	100 (36.5-100)	82.2 (80.2-83.9)	83.1 (18.0-100)	95.2 (20.6-100)	60.1 (57.9-62.2)	36.1 (8.9-100)	45.0 (9.3-100)	26.3 (24.4-28.2)
Every 8-week DP									
Clinical trial protocol	20.8 (6.2-81.2)	28.9 (7.8-97.0)	6.7 (5.6-7.9)	11.9 (3.9-57.0)	15.8 (5.0-74.1)	1.7 (1.2-2.2)	6.3 (1.8-26.3)	7.9 (2.4-39.1)	.1 (<.01-.3)
WHO 2015	27.6	46.6	16.2	15.3	26.7	5.5	7.7	11.9	.7

	(7.6-93.2)	(11.5-100)	(14.5-17.9)	(4.9-70.6)	(7.3-94.7)	(4.5-6.5)	(2.3-37.2)	(3.5-62.1)	(.4-1.0)
Proposed age-based	44.4 (11.6-100)	51.9 (12.9-100)	19.9 (17.9-21.8)	25.7 (7.3-89.1)	30.0 (8.1-96.6)	7.0 (5.9-8.2)	11.7 (3.5-57.2)	13.2 (3.9-65.6)	.9 (.5-1.3)

WAZ = weight-for-age z-score

Clinical trial protocol: <6 kg: DHA/PPQ 10/80 mg daily x 3 days, 6-<11 kg: DHA/PPQ 20/160 mg daily x 3 days, 11-<15 kg:

DHA/PPQ 30/240 mg daily x 3 days, 15-<20 kg: DHA/PPQ 40/320 mg daily x 3 days

WHO 2015: <8 kg: DHA/PPQ 20/160 mg daily x 3 days, 8-<11 kg: DHA/PPQ 30/240 mg daily x 3 days, 11-<17 kg: DHA/PPQ 40/320 mg daily x 3 days, 17-<25 kg: DHA/PPQ 50/480 mg daily x 3 days

Proposed age-based: 2-6 months of age: DHA/PPQ 20/160 mg daily x 3 days, 6-18 months of age: DHA/PPQ 30/240 mg daily x 3 days, 18-24 months of age: DHA/PPQ 40/320 mg daily x 3 days

Figure 6. Simulation results. (A) Predicted PPQ trough concentrations for simulated DP regimens, stratified by nutritional status. The red line indicates the 15.4 ng/mL PPQ target. Boxes indicate 25-75% of population incidence, and vertical bars represent 95% of the population. (B) Predicted malaria incidence by DP regimen with increasing baseline malaria transmission, stratified by nutritional status and adherence level (1/3 adherence indicates bioavailability observed for non-directly observed therapy in the study, 2/3 adherence indicates a bioavailability midpoint between the directly and non-directly observed population, and full adherence indicates the bioavailability observed in the directly observed therapy group).

Table 2: It would increase the interpretation of the data by presenting IIV as %CV, and to add RSE (from the bootstrap).

Response: We have added these to table 2 as shown below.

Table 2. Pharmacokinetic and pharmacodynamic parameters

Parameter	Value (%RSE, 95% CI)	Interindividual Variability, % (%RSE, 95% CI)
Pharmacokinetic parameters		
N (subjects/PK observations)	280/4573	
Clearance (L/d) [†]	435 (6.1%, 385-492)	27.4 (15.9%, 22.0-32.0)
Θ_{Age} [‡]	.269 (7.0%, .231-.304)	-
Volume of central compartment (L)	595 (10.6%, 489-733)	32.8 (37.6%, 11.3-47.5)
Intercompartmental clearance 1 (L/d)	517 (12.2%, 405-657)	-
Volume of peripheral compartment 1 (L)	7310 (8.4%, 6270-8650)	-
Intercompartmental clearance 2 (L/d)	674 (16.8%, 466-934)	-
Volume of peripheral compartment 2 (L)	1060 (18.9%, 702-1490)	-
Absorption Compartments [*]	2	-
Absorption Transit Time (d)	.041 (9.4%, .033-.049)	43.2 (22.0-60.6)
$\Theta_{\text{Capillary to venous conversion}}$ [†]	0.922	-
Proportional error	44.6% (1.6%, 43.2%-46.1%)	-
Relative bioavailability (F) [*]	1	-
$\Theta_{\text{Weight for age z-score}}$ ^{**}	.105 (21.6%, .050-.135)	-
$\Theta_{\text{Self-administered therapy}}$ ^{**}	.389 (7.3%, .339-.452)	-
$\Theta_{\text{Between occasion variability}}$	66.7% (5.9%, 61.3%-71.1%)	-
Pharmacodynamic parameters		
N (subjects/observations)	280/326	
Baseline hazard/1000 (per day) ^{***}	.398 (17.8%,.265-.536)	69.4 (62.0, 45.2-147)
Season adjustment		
$\Theta_{\text{Transmission season 2015}}$	1.30 (34.8%, 0.67-2.45)	-
$\Theta_{\text{Transmission season 2016}}$	5.25 (17.6%, 3.81-7.54)	-
$\Theta_{\text{Transmission season 2017}}$	7.93 (17.7%, 5.73-	-

$\Theta_{PPQ\ EC50}$ (ng/mL)	5.97 (14.8%, 4.19-7.60)	-
$\Theta_{PPQ\ \gamma}$	3.10 (24,0%, 1.98-4.79)	

Confidence intervals obtained by bootstrap (n=1,000)

* Prespecified

† $CL = \text{Population CL} * \left(\frac{\text{Weight}}{8.6\text{ kg}}\right)^{\theta.75} * \left(\frac{\text{Age}}{57\text{ weeks}}\right)^{\theta}$; normalized for 8.6 kg child at 57 weeks of age

† $\ln([PPQ]_{\text{capillary}}) = \Theta_{\text{slope}} * \ln([PPQ]_{\text{venous}})$

** $F = 1 * \Theta_{\text{Self-administered therapy}} * (1 + \Theta * (\text{WAZ} - (-.5))) * e^{\Theta_{\text{Between occasion variability}}}$, WAZ=weight for age z-score

*** Survival function: $\Theta_{\text{Baseline}}/1000 * \Theta_{\text{Transmission season}} * \frac{[PPQ]^{\gamma}}{\Theta_{\text{EC50}}^{\gamma} + [PPQ]^{\gamma}}$

Figure S4: change y-axis to mSec

Response: This has been changed as shown below.

Reviewers' Comments:

Reviewer #2:

Remarks to the Author:

All reviewer comments have been adequately addressed.

Reviewer #3:

Remarks to the Author:

My comments have been answered satisfactory, but I do disagree with the authors conclusion regarding the enzyme maturation model. If the power function model showed a similar shape as the Emax-type model during the observable data range (as stated), I would always choose the biologically plausible model (Emax model), irrespective of OFV. This would enable you and other researchers to extrapolate beyond the observed data for translational simulations. I leave it to the authors to make a decision for the final publication.

Overall, a great paper and I congratulate the authors on their work.

Prof. Joel Tarning

Reviewer #3:

My comments have been answered satisfactory, but I do disagree with the authors conclusion regarding the enzyme maturation model. If the power function model showed a similar shape as the Emax-type model during the observable data range (as stated), I would always choose the biologically plausible model (Emax model), irrespective of OFV. This would enable you and other researchers to extrapolate beyond the observed data for translational simulations. I leave it to the authors to make a decision for the final publication.

Response: Thank you for this comment, and we have revised the pharmacokinetic model to include the Emax relationship for enzyme maturation as recommended by the reviewer. Fortunately, this was a minor change and did not lead to substantial changes to our PK/PD model estimates, results or conclusions.

We made the following change to the manuscript:

- “PPQ is primarily metabolized by cytochrome P450 3A4 (7), and a maturation function using post menstrual age (PMA) to represent liver enzyme maturation during early childhood significantly modified PPQ clearance as shown in equation 1, where θ_{CL} is the population clearance, θ_{EC50} is the PMA when maturation reaches 50%, and η is the between subject random term (Δ OFV -331; Table 2).

$$CL = \theta_{CL} * \left(\frac{WEIGHT}{8.6 \text{ kg}}\right)^{0.75} * \left(\frac{PMA}{PMA + \theta_{EC50}}\right) * e^{\eta} \quad (1)'' \text{ (page 6, line 112)}$$

- We revised the model parameter estimates and confidence intervals in Table 2 to reflect the updated models.
- We revised the simulation results in Table 3 using the new model, and the new values extremely similar to the prior results.
- We revised Figure 6 to reflect the simulations using the updated PK model, and again changes were minimal and our conclusions remain the same.